# ☕ AGENTORCHESTRA: ORCHESTRATING HIERARCHICAL MULTI-AGENT INTELLIGENCE WITH THE TOOL-ENVIRONMENT-AGENT(TEA) PROTOCOL

## ABSTRACT

Recent advances in LLMs-based agent systems have demonstrated remarkable capabilities in solving complex tasks. Nevertheless, current protocols (e.g., A2A and MCP) suffer from insufficient capabilities in context management, limited adaptability to diverse environments, and the absence of dynamic agent architectures. To address these limitations, we propose the **Tool-Environment-Agent** (TEA) Protocol, which establishes a principled basis for integrating environments, agents, and tools into an unified system. The TEA protocol treats environments and agents as first-class resources, enabling comprehensive context management and adaptive environment integration. Based on this protocol, we introduce **AGENTORCHESTRA**, a hierarchical multi-agent framework with a central planning agent that decomposes complex objectives and coordinates specialized agents. Each sub-agent is dedicated to specific functions, providing capabilities for data analysis, file operations, web navigation, and interactive reasoning. Notably, **AGENTORCHESTRA** introduces a tool manager agent that supports intelligent evolution through dynamic tool creation, retrieval, and reuse mechanisms. Experiments on three widely used benchmarks show that **AGENTORCHESTRA** consistently outperforms existing baselines, achieving state-of-the-art performance of 83.39% on GAIA and ranking among the top general-purpose LLM-based agents. These results highlight the effectiveness of the TEA Protocol and hierarchical organization in building general-purpose multi-agent systems.

## 1 INTRODUCTION

Recent advances in LLMs-based agent systems have demonstrated remarkable capabilities in solving both general-purpose and highly complex tasks across various domains, including web browsing (OpenAI, 2025b; Müller & Žunič, 2024), computer operation (Anthropic, 2024a; Qin et al., 2025), code execution (Wang et al., 2024a), game playing (Wang et al., 2023; Tan et al., 2024), and research assistance (OpenAI, 2024; Deep-Mind, 2024; xAI, 2025). However, current foundation agents still struggle to generalize across different scenarios, primarily due to the dramatic differences in environment encapsulation methods and the reliance on manually designed observation-action spaces.

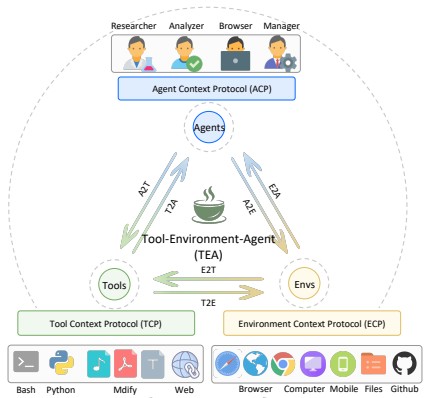

Figure 1: Overview of the TEA Protocol.

Additionally, current agent protocols face significant limitations that hinder their ability to serve as universal solutions for general-purpose tasks. Existing protocols such as Google's Agent2Agent (A2A) (Google, 2025) and Anthropic's Model Context Protocol (MCP) (Anthropic, 2024b) suffer from three fundamental issues: i) **Insufficient capabilities in context management** that fail to capture the full complexity and context of available resources, limiting effective tool selection and utilization; ii) **Inability to adapt to arbitrary environments**,

where environment encapsulation methods vary drastically and observation-action spaces largely rely on manual design, constraining their effectiveness in complex, multi-domain scenarios; and iii) **Lack of dynamic agent architecture**, which rigidly defines agents as fixed and pre-determined structures, thereby limiting their capacity to function as adaptive collaborators and hindering the emergence of coordinated and flexible agent behaviors in complex task scenarios.

To address these fundamental limitations, we propose the **Tool-Environment-Agent** (TEA) Protocol, a unified protocol that seamlessly integrates environments, agents, and tools into a cohesive system, as illustrated in Figure 1. The TEA Protocol extends beyond traditional tool-based approaches by treating environments and agents as first-class resources, enabling comprehensive context management and adaptive environment integration through a standardized interface that unifies diverse computational resources. This design allows agents to directly access and control environments, invoke other agents, and utilize tools through a consistent and standardized protocol, thereby eliminating the need for environment-specific adaptations, manual interface design, and redundant integration efforts. As simple as brewing tea, the TEA Protocol makes building agents a graceful, harmonious experience that unlocks infinite possibilities for collaboration and intelligence.

> *"Some people will tell you there is a great deal of poetry and fine sentiment in a chest of tea."*
>
> — Ralph Waldo Emerson

Building upon this foundation, we introduce **AGENTORCHESTRA**, a hierarchical multi-agent framework for general-purpose task solving that integrates high-level planning with modular agent collaboration. **AGENTORCHESTRA** features a central planning agent that decomposes complex objectives and delegates sub-tasks to a team of specialized agents, including deep researcher agent, browser use agent, deep analyzer agent, and tool manager agent, each equipped with domain-specific environments and tools. Our contributions are threefold:

- We propose the TEA Protocol, a unified framework that seamlessly integrates environments, agents, and tools, addressing the fundamental limitations of existing protocols.
- We present **AGENTORCHESTRA** as an instance application of the TEA Protocol, designed as a hierarchical multi-agent framework that demonstrates the protocol's practicality and effectiveness in real-world scenarios.
- Extensive experiments demonstrate the effectiveness of both the TEA Protocol and **AGENTORCHESTRA**, which consistently outperforms existing agent baselines, achieving state-of-the-art performance 83.39% on GAIA benchmark, ranking among the top general-purpose agents.

## 2 RELATED WORK

### 2.1 TOOL AND AGENT PROTOCOLS

Recent protocols have focused on standardizing tool interfaces and agent communication. For instance, MCP (Anthropic, 2024b) unifies tool integration for LLMs agents, while A2A protocol (Google, 2025) enables agent-to-agent messaging and coordination. Other efforts, such as the Agent Communication Protocol (ACP) (Ehtesham et al., 2025), the Agent Network Protocol (ANP) (Ehtesham et al., 2025), and frameworks like SAFEFLOW (Li et al., 2025), further enhance interoperability, discovery, and safety in multi-agent systems. However, these approaches predominantly treat agents and tools as isolated or static components, overlooking environments as dynamic, first-class resources, which limits adaptive orchestration and richer collaboration.

### 2.2 GENEGENERAL-PURPOSE AGENTS

The integration of tools with LLMs marks a paradigm shift in AI agent development, with tool-augmented LLM agents exhibiting greater flexibility, cross-domain reasoning, and natural language interaction (Liang & Tong, 2025). These agents have demonstrated strong capabilities in web browsing (OpenAI, 2025b; Müller & Žunič, 2024), computer operation (Anthropic, 2024a; Qin et al., 2025), code execution (Wang et al., 2024a), and game playing (Wang et al., 2023; Tan et al., 2024). Standardized tool interfaces, such as OpenAI's Function Calling and Anthropic's MCP, have further streamlined tool integration (OpenAI, 2023; Anthropic, 2024b), while frameworks like ToolMaker (Wölflein et al., 2025) enable automatic transformation of code-based research into

LLM-compatible tools. Building upon these foundations, multi-agent systems have seen substantial growth, with systems like MetaGPT (Hong et al., 2023) demonstrating how specialized agents can coordinate to solve complex problems beyond single agents' reach. Recent work by Li et al. (Li et al., 2024) and Ni et al. (Ni et al., 2025) has further advanced collaborative reasoning and self-improving social agent frameworks. Nevertheless, many existing approaches still lack mechanisms for efficient communication, dynamic role allocation, and coordinated teamwork in large-scale tasks. The rise of generalist agents and open-source frameworks, such as Manus (Shen & Yang, 2025), OpenHands (Wang et al., 2024b), OpenManus (Liang et al., 2025), and smolagents (Roucher et al., 2025), has advanced unified perception, reasoning, and tool-augmented action beyond domain-specific applications. Recent work like Alita (Qiu et al., 2025) introduces novel approaches to generalist agents through minimal predefinition and maximal self-evolution, while comprehensive surveys (Lu & Wang, 2020) document the evolution from task-specific agents to more flexible, general-purpose systems. However, these agents and frameworks lack unified protocols and have limited general-purpose capabilities, which motivates us to propose the TEA Protocol and build a general-purpose multi-agent framework based on it.

## 3  THE TEA PROTOCOL

Before introducing our concrete implementation **AGEN-TORCHESTRA**, we first present the TEA Protocol, as illustrated in Figures 1 and 2. The TEA Protocol consists of three main components: 1) **Infrastructure Layer** defines the foundational components, including the unified interface for LLM models and the memory system; 2) **Core Protocols** that separately define the Tool Context Protocol (TCP), Environment Context Protocol (ECP), and Agent Context Protocol (ACP) for managing tools, environments, and agents respectively; and 3) **Protocol Transformations** that define the inter-conversion relationships between TCP, ECP, and ACP, enabling seamless resource orchestration and dynamic

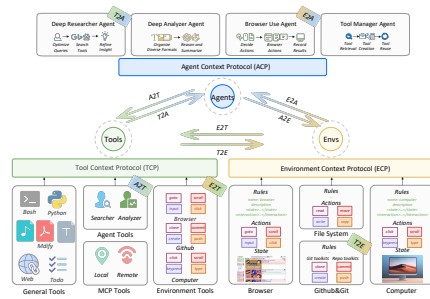

Figure 2: Architecture of the TEA Protocol.

adaptation across different entities. Details and formalization can be found in Appendix C.

**Definition 1** (TEA Protocol). *Let $\mathcal{T}, \mathcal{E}, \mathcal{A}$ be sets of tools, environments, and agents, with*

$$T = \langle \mathcal{I}_T, \mathcal{O}_T, \phi_T \rangle, \quad E = \langle \mathcal{S}_E, \mathcal{A}_E, \tau_E \rangle, \quad A = \langle \mathcal{X}_A, \mathcal{A}_A, \pi_A \rangle.$$

*The TEA protocol is*

$$\text{TEA} = \langle \mathcal{T}, \mathcal{E}, \mathcal{A}, \Sigma, \mathcal{C}, \mathcal{P} \rangle, \quad \mathcal{P} = \{\text{A2T, E2T, T2E, T2A, A2E, E2A}\},$$

*where $\Sigma$ is a metadata/relations registry, $\mathcal{C}$ a context binder, and $\mathcal{P}$ is the family of cross-domain transformations.*

### 3.1  INFRASTRUCTURE LAYER

The Infrastructure Layer provides the foundational components of the TEA Protocol, including a unified interface for diverse LLMs (e.g., `gpt-5`) that abstracts model heterogeneity, and an integrated memory system for persistent contextual storage and knowledge management across sessions.

### 3.2  CORE PROTOCOLS

**Tool Context Protocol.** MCP (Anthropic, 2024b) is the most widely adopted tool protocol, defined by three components: tools, prompts, and resources. However, MCP suffers from several limitations: i) Inadequate parameter descriptions make it difficult for LLMs to provide appropriate parameters; ii) Lack of tool relationship modeling prevents describing associations between tools; and iii) Absence of context management constrains coherence across tool use.

To address these limitations, we propose the **Tool Context Protocol** (TCP), which extends MCP by supporting local and remote tool loading, detailed tool registration, and the novel ability to register agents as tools for dynamic transformations. Additionally, TCP represents environment-provided toolkits as contextually described tool collections, providing rich semantic information about tool relationships and environmental constraints. Moreover, TCP stores each tool with an embedding and

uses query–embedding similarity for candidate retrieval to improve selection efficiency through its tool context manager that controls tool lifecycle and execution context.

**Environment Context Protocol.** In reinforcement learning, frameworks such as Gym (Brockman et al., 2016) provide standardized interfaces for training and testing environments. However, most existing research on general-purpose agent systems either focuses on single environments or relies on ad-hoc adaptations, seldom addressing unified environment interfaces. Recent attempts to encapsulate environments as MCP tools allow agent interaction, but lack mechanisms to capture inter-tool dependencies and manage contextual execution environments.

To overcome these limitations, we introduce the **Environment Context Protocol** (ECP), a flexible protocol that defines unified inputs, outputs, and environment rules across multiple environments. ECP registers the environment name, description and environment-specific usage rules (e.g., browser for web navigation operations), then incorporates the entire action space into a toolkit, enabling agents to invoke actions as contextually informed tools through its environment context manager that maintains environment state and execution context. This design facilitates seamless integration of heterogeneous environments and supports adaptive context management across diverse domains.

**Agent Context Protocol.** Existing agent frameworks (Roucher et al., 2025; Liang et al., 2025) typically rely on ad-hoc strategies for defining and managing agents. Each agent is associated with specific roles, capabilities, and policies. However, such systems often exhibit poor interoperability and lack standardized representations of agent attributes. Furthermore, they provide insufficient means to capture inter-agent interactions such as delegation, collaboration, or hierarchical organization. Most current approaches also fail to explicitly encode the contextual environments in which agents operate. This limitation complicates consistent state maintenance in multi-agent scenarios.

To address these limitations, we propose the **Agent Context Protocol (ACP)**. At its core, ACP incorporates an agent context manager that maintains agent states and execution contexts, providing a foundation for persistent coordination. Building on this foundation, ACP establishes a unified schema for registering, representing, and orchestrating agents within the TEA Protocol. It supports semantically enriched metadata to capture agents' roles, competencies, and objectives, while enabling state persistence across tasks and sessions. Furthermore, ACP formalizes the modeling of inter-agent dynamics, supporting cooperative, competitive, and hierarchical configurations. By embedding contextualized descriptions of agents and their interactions, ACP facilitates flexible orchestration, adaptive collaboration, and systematic integration with TCP and ECP.

### 3.3 PROTOCOL TRANSFORMATIONS

While TCP, ECP, and ACP provide independent specifications for tools, environments, and agents, practical deployment requires interoperability across these protocols. Real-world scenarios often demand that entities assume alternative roles or exchange contextual information in a principled manner. For example, an environment originally serving as a static resource set may need to be encapsulated into a toolkit for agent interaction, while tools with fixed functions may need to be enhanced into intelligent systems capable of complex reasoning or autonomous task execution to support more advanced workflows. These transformations are essential for dynamic resource orchestration, allowing computational entities to adapt their functional scope to evolving task demands and system constraints. To this end, we identify six fundamental categories of protocol transformations:

- **Agent-to-Tool** (A2T). Encapsulates an agent's capabilities and reasoning into a standardized tool interface, enabling seamless integration with existing tool ecosystems. For example, a deep researcher workflow can be instantiated as a tool for internet-scale retrieval tasks.

- **Tool-to-Agent** (T2A). Designates tools as an agent's actuators, translating goals into parameterized invocations. For instance, a data analysis agent may use SQL tools to query databases, while a design agent may apply image editing tools for creative modifications.

- **Environment-to-Tool** (E2T). Converts environment-specific actions into standardized interfaces, allowing agents to interact via consistent tool calls. For example, unifying browser actions like Navigate, GoBack, and Click into a context-aware toolkit.

- **Tool-to-Environment** (T2E). Elevates a tool set into an environment abstraction, treating individual functions as actions within a unified action space. For instance, code editing, compilation, and debugging tools can be encapsulated as a programming environment.

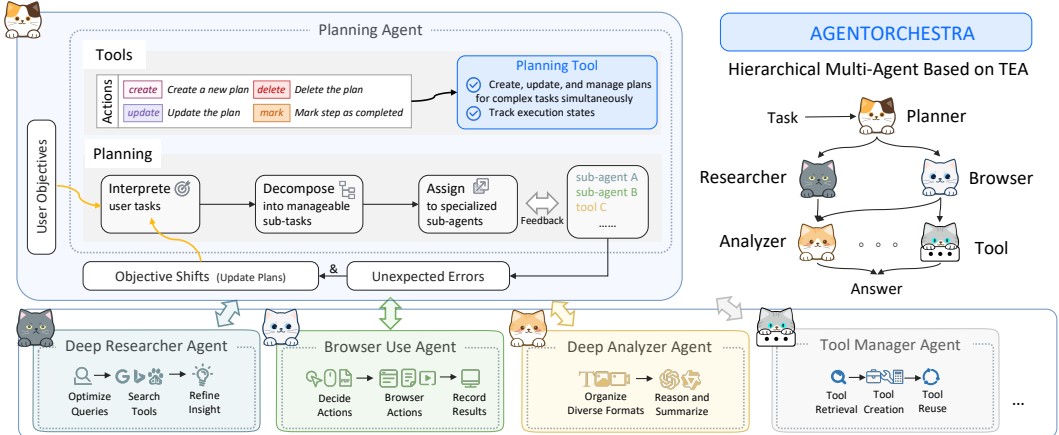

Figure 3: Architecture of **AGENTORCHESTRA**.

- **Agent-to-Environment** (A2E). Encapsulates an existing agent as an interactive environment, exposing its decision rules and behavioral dynamics for other agents to explore, learn, or be evaluated. For example, a trained trading agent can be turned into a market simulation for testing new trading strategies.

- **Environment-to-Agent** (E2A). Infuses reasoning and adaptive decision-making into an environment's state dynamics, transforming it into an autonomous agent capable of pursuing goals and interacting strategically. For instance, a game environment can evolve into an AI opponent that adapts its strategy to player behavior.

These six transformation categories establish a comprehensive framework for dynamic resource orchestration within the TEA Protocol. By enabling seamless transitions between tools, environments, and agents, the protocol transformations support adaptive architectures that reconfigure functional components in response to task requirements and contextual constraints.

## 4 AGENTORCHESTRA

To validate the TEA Protocol, we implement **AGENTORCHESTRA**, a hierarchical multi-agent framework for generalization, multimodal reasoning, scalability, and collaboration. It employs a two-tier design: a planning agent decomposes tasks and coordinates sub-agents, enabling flexible composition and scalable adaptation. Section 4.1 introduces the core design principles of this framework. Section 4.2 details the implementation of the planning agent, and Section 4.3 discusses the architecture and interaction patterns of specialized sub-agents. Details can be found at Appendix E.

### 4.1 AGENT DESIGN PRINCIPLES

Within the TEA Protocol framework, six key entities are defined. An **agent** is an autonomous computational entity that perceives, interprets, and flexibly acts across diverse tasks. The **environment** represents the external context and resources within which the agent operates, standardized by the ECP. A **model**, typically an LLM, provides reasoning and decision-making capabilities, with the Infrastructure Layer enabling dynamic selection across different models. **Memory** persistently records execution histories, automatically summarizing and extracting insights to assist task completion. An **observation** captures task descriptions, execution histories, environment states, and tool availability, providing a comprehensive view for the agent. Finally, an **action** is managed through the TCP and executed via parameterized tool interfaces. Details can be found in Appendix D.

An agent operates in a perception–interpretation–action cycle. It observes the environment and stores information in memory, interprets context with the unified LLMs interface, and determines an action. The action is executed in a sandbox, with results recorded back to memory to refine reasoning and adaptation. This loop continues until objectives are achieved or a termination condition is met.

## 4.2 PLANNING AGENT

The planning agent serves as the central orchestrator in our hierarchical framework, dedicated to high-level reasoning, task decomposition, and adaptive planning. It interprets user objectives and systematically decomposes complex tasks into manageable sub-tasks, which are assigned to specialized sub-agents or tools based on their expertise. The planning agent maintains a global perspective throughout execution, aggregating feedback and monitoring progress toward the overall objective. This enables dynamic plan updates, adapting strategy in real time in response to intermediate results, unexpected challenges, or shifting user requirements. To ensure modularity and scalability, the planning agent interacts with sub-agents through the ACP and utilizes tools from the TCP, concealing domain-specific details and facilitating the integration of new agent types and resources.

The planning agent is implemented as a React-based (Yao et al., 2023) tool-calling agent that follows a systematic thinking-then-action paradigm, as detailed in Section I. During execution, it records its decision-making process and trajectory in memory, continuously summarizing and extracting insights from experience, and employs a done tool to determine task completion, ensuring reliable termination of complex workflows. A dedicated todo tool supports task decomposition and step tracking, where each task is a structured step with attributes such as identifier, description, parameters, priority, category, status, and result. The todo tool enables adding, updating, completing, listing, clearing, and exporting steps, while synchronizing changes between an internal step list and a human-readable todo.md file. Planning granularity is defined at the sub-task level, with each sub-task executable by a specialized sub-agent or a composition of tools, enabling persistent and interpretable workflow management that complements high-level reasoning with fine-grained progress monitoring. To improve efficiency, specialized sub-agents are designed as lightweight custom workflows that avoid the extensive system prompt overhead of the planning agent, balancing task completion performance with reduced token consumption.

## 4.3 SPECIALIZED SUB-AGENTS

To address real-world challenges such as comprehensive information retrieval, domain-specific expertise acquisition, statistical analysis, and computational tasks, we instantiate our hierarchical multi-agent framework with specialized sub-agents for distinct task stages. A deep researcher agent conducts large-scale information retrieval by efficiently scanning and filtering web pages to identify promising sources. A browser use agent enables fine-grained interaction with web content, directly engaging with videos, pdfs, and html elements to extract precise information. A deep analyzer agent performs advanced reasoning and integrative analysis, leveraging collected data for tasks such as statistical inference, image analysis, and market studies. A tool manager agent enables intelligent tool evolution through automated creation, dynamic retrieval, and systematic reuse of tools, allowing the system to autonomously extend its capabilities. Each sub-agent is equipped with a specialized python interpreter for data analysis and self-checking via code-based reasoning.

### 4.3.1 DEEP RESEARCHER AGENT

The deep researcher agent is a specialized module for comprehensive information gathering, implemented as a multi-round, multimodal research workflow. Inspired by OpenManus Liang et al. (2025), it follows a query-driven paradigm: given a research task with text or image inputs, the agent generates optimized search queries using LLM prompts, performs breadth-first searches across multiple engines (e.g., Google, Bing, Firecrawl), fetches and analyzes web content, extracts key insights, and recursively issues follow-up queries until sufficient information is collected or a predefined limit is reached. Its multimodal support enables simultaneous processing of text and visual data, improving understanding of complex contexts and extraction of relevant insights. All visited URLs, extracted information, and generated queries are stored in a structured research history, culminating in a relevance-ranked, source-cited summary that supports transparent and scalable knowledge synthesis.

### 4.3.2 BROWSER USE AGENT

The browser use agent is a specialized agent for automated and fine-grained web interaction, designed to complement the exploratory focus of the deep researcher agent with precise, task-oriented information acquisition. Implemented under the ECP protocol, it first provides a playwright-based browser environment and then leverages the E2T transformation to supply a browser interaction

toolkit, enabling the agent to perform a wide spectrum of web operations. These include search, navigation, content extraction, document manipulation, dynamic form filling, PDF and video control, as well as robust tab and session management. Through its action-based design, the agent maintains fine-grained execution control and extensibility for integrating new web operations.

Certain tasks, such as Google Street View navigation, interactive maps, 3D visualizations, and multimedia applications, cannot be effectively handled through DOM-level control alone, as they require pixel-level operations (e.g., precise mouse movements, drag-and-drop, and keyboard events). The ECP Protocol provides a key advantage here: by seamlessly integrating both browser and computer environments, the browser use agent can access and alternate between the two toolkits, achieving unified control across DOM-based and pixel-level interactions. This integration enables the agent to perform sophisticated hybrid workflows that combine web automation with low-level computer operations, thereby expanding its capacity to handle previously inaccessible interactive elements and complex real-world tasks.

### 4.3.3 DEEP ANALYZER AGENT

The deep analyzer agent is a workflow-oriented agent for multi-step analysis of complex reasoning tasks with diverse data sources. It supports a wide range of file formats including text, code, documents, images, audio, and video, and integrates multimodal inputs into the reasoning process. For each task, it organizes materials into an enhanced context, performs iterative analysis to extract insights, and synthesizes results into coherent conclusions. Analysis steps are recorded for transparency, and adaptive evaluation determines task completeness. Final outputs are structured reports containing summaries, key findings, and recommendations, while its extensible design ensures adaptability to new data modalities and evolving analytical requirements.

### 4.3.4 TOOL MANAGER AGENT

The rapid expansion of AI agent applications has led to an exponential growth in the complexity and diversity of required tools, encompassing code generation, data querying, formatting operations, and domain-specific functionalities. Traditional approaches relying on manual tool development and maintenance face significant challenges, including development inefficiency, version inconsistency, and limited adaptability to emerging requirements. To address these limitations, we introduce the tool manager agent, a specialized component managed under the TCP that enables intelligent tool evolution through automated creation, dynamic retrieval, and systematic reuse mechanisms. This agent can either store tools as ordinary components within the TCP or expose them as MCP-style servers to provide remote agents with access to these capabilities, marking a paradigm shift from static tool provisioning to adaptive tool ecosystem management.

The tool manager agent is designed around three core principles: tool retrieval, tool creation, and tool reuse. To address the challenge of continuously growing TCP tool libraries and the limited concurrent invocation capacity of mainstream function-calling-based LLMs, the agent employs a keyword-based pre-filtering strategy to efficiently select candidate tool subsets for decision-making, while triggering automatic creation when no suitable tools are available. The tool creation process follows a systematic methodology comprising intent analysis, code synthesis, validation, and registration, enabling the generation of TCP-compliant tools with standardized definitions, robust error-handling mechanisms, and performance optimization. Validated tools are directly registered into the TCP framework, thereby integrating them into the unified management and scheduling system. This generate–validate–register–reuse loop establishes a scalable and adaptive tool management ecosystem, ensuring consistency, efficiency, and extensibility in large-scale agent deployments.

## 5 EMPIRICAL STUDIES

This section presents our experimental setup and results, including benchmark evaluations, baseline comparisons, and comprehensive analysis. Additional examples are provided in the Appendix G.

**Experimental Settings**. We evaluate our framework on three benchmarks: **SimpleQA** Wei et al. (2024), a 4,326-question factual accuracy benchmark; **GAIA** Mialon et al. (2023), assessing real-world reasoning, multimodal processing, and tool use with 301 test and 165 validation questions; and **Humanity's Last Exam (HLE)** Phan et al. (2025), a 2,500-question multimodal benchmark

for human-level reasoning and general intelligence. We report score (pass@1), which measures the proportion of questions for which the top prediction is fully correct. Specifically, the planning agent ($m$=20), deep researcher ($m$=3), and tool manager ($m$=10) are built on `claude-3.7-sonnet`; the browser agent uses `gpt-4.1` ($m$=5) and `computer-use-preview(4o)` ($m$=50); and the deep analyzer employs `gemini-2.5-pro` and `o3` ($m$=3), where $m$ denotes the maximum steps.

## 5.1 Performance across Benchmarks

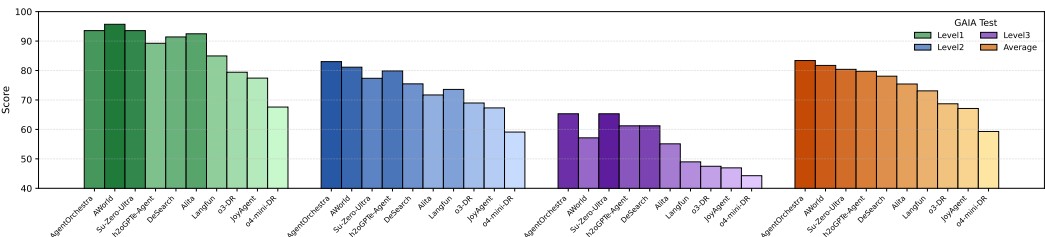

Figure 4: GAIA Test Results.

**GAIA.** Our **AGENTORCHESTRA** achieves SOTA results with 83.39% overall accuracy, representing a 4% improvement over the baseline without tool manager agent (79.07%). The system demonstrates strong performance across all difficulty levels (92.45% Level 1, 83.72% Level 2, 57.69% Level 3), consistently outperforming advanced baselines such as AWORLD (77.58%) and Langfun Agent (76.97%). The planning agent orchestrates task decomposition through dynamic routing to specialized agents. The browser use agent leverages ECP-based environment integration for precise web data extraction, while the deep analyzer agent employs structured workflows for multimodal reasoning. The tool manager agent autonomously generates context-specific tools through TCP-based management, excelling in structured data retrieval but facing challenges in fine-grained visual analysis.

Table 1: Performance on GAIA Validation.

| Agents | Level 1 | Level 2 | Level 3 | Average |
|---|---|---|---|---|
| HF ODR (o1) (HuggingFace, 2024) | 67.92 | 53.49 | 34.62 | 55.15 |
| OpenAI DR (OpenAI, 2024) | 74.29 | 69.06 | 47.60 | 67.36 |
| Manus (Shen & Yang, 2025) | 86.50 | 70.10 | 57.69 | 73.90 |
| Langfun (Google, 2024) | 86.79 | 76.74 | 57.69 | 76.97 |
| AWorld (Yu et al., 2025) | 88.68 | 77.91 | 53.85 | 77.58 |
| **AGENTORCHESTRA** | **92.45** | **83.72** | **57.69** | **82.42** |

**SimpleQA.** Our **AGENTORCHESTRA** achieves SOTA performance with 95.3% accuracy, substantially outperforming leading LLM baselines such as o3 (49.4%) and gemini-2.5-pro (50.8%), and surpassing strong agent-based baselines including Perplexity Deep Research (93.9%). The system excels in factoid question answering through systematic cross-verification mechanisms, where multiple information sources are retrieved and validated to ensure answer accuracy. This multi-source validation approach substantially reduces hallucination risks by grounding responses in verified information, demonstrating the effectiveness of hierarchical agent coordination for knowledge-intensive tasks requiring high factual accuracy.

Table 2: Performance on SimpleQA and HLE.

| Model and Agent | SimpleQA |
|---|---|
| **Models** | |
| o3 (w/o tools) | 49.4 |
| gemini-2.5-pro-preview-05-06 | 50.8 |
| **Agents** | |
| Perplexity DR (Perplexity, 2025) | 93.9 |
| **AGENTORCHESTRA** | **95.3** |

| Model and Agent | HLE |
|---|---|
| **Models** | |
| o3 (w/o tools) | 20.3 |
| claude-3.7-sonnet (w/o tools) | 8.9 |
| gemini-2.5-pro-preview-05-06 | 17.8 |
| **Agents** | |
| OpenAI DR (OpenAI, 2024) | **26.6** |
| Perplexity DR (Perplexity, 2025) | 21.1 |
| **AGENTORCHESTRA** | 25.9 |

**HLE.** Our system achieves 25.9% on the HLE benchmark, surpassing baselines such as o3 (20.3%), gemini-2.5-pro (17.8%), claude-3.7-sonnet (8.9%), and Perplexity Deep Research (21.1%). The system demonstrates superior performance in high-level reasoning tasks requiring sustained analytical thinking and expert knowledge integration. The hierarchical architecture enables complex problem decomposition and multi-step reasoning,

where specialized agents tackle different aspects of challenging problems while maintaining coherent solution pathways.

## 5.2 ABLATION STUDIES

We mainly conducted ablation studies on the GAIA Test to verify the effectiveness of each sub-agent in **AGENTORCHESTRA**, as well as the reuse rate of the new tools created by the tool manager agent.

**Effectiveness of the specialized sub-agents**. We conduct ablation studies to evaluate the contribution of each specialized sub-agent in **AGENTORCHESTRA**, where P, R, B, A, and T represent the planning agent, deep researcher agent, browser use agent, deep analyzer agent, and tool manager agent, respectively.

Table 3: Sub-agent effectiveness across GAIA Test.

| P | R | B | A | T | Level 1 | Level 2 | Level 3 | Average | Improvement |
|---|---|---|---|---|---------|---------|---------|---------|-------------|
| ✓ | | | | | 54.84 | 33.96 | 10.20 | 36.54 | – |
| ✓ | ✓ | | | | 86.02 | 47.17 | 34.69 | 57.14 | +56.40% |
| ✓ | ✓ | ✓ | | | 89.25 | 71.07 | 46.94 | 72.76 | +27.33% |
| ✓ | ✓ | ✓ | ✓ | | 91.40 | 77.36 | 61.22 | 79.07 | +8.67% |
| ✓ | ✓ | ✓ | ✓ | ✓ | 93.55 | 83.02 | 65.31 | 83.39 | +5.46% |

The GAIA benchmark contains over 350 questions requiring network information retrieval, making it ideal for evaluating multi-agent coordination. When equipped with both coarse-grained retrieval (deep researcher agent) and fine-grained web interaction (browser use agent), performance nearly doubles from 36.54% to 72.76%. The deep analyzer agent contributes an additional 8% improvement for complex reasoning tasks, while the tool manager agent provides a final 5% boost through adaptive tool generation. These results demonstrate the critical importance of specialized agent coordination for comprehensive task-solving capabilities.

**Reuse rate of the created tools**. The tool manager agent demonstrates efficient tool creation and reuse capabilities, generating over 50 tools during evaluation with a 30% reuse rate. This indicates an effective balance between tool specialization for specific tasks and generalization for broader applicability, contributing to the system's adaptability and resource efficiency.

## 6 LIMITATIONS AND FUTURE WORK

Despite TEA being a highly compatible protocol and **AGENTORCHESTRA** being a general-purpose agent implemented based on TEA, several limitations remain. First, TEA currently does not support dynamic agent role allocation, enabling automatic role assignment during multi-agent runtime. Additionally, the TEA protocol does not yet support agent self-evolution, such as dynamic optimization of prompts, tools, and agent structures during runtime. Second, while **AGENTORCHESTRA** demonstrates promising potential in tool evolution, it still faces challenges in handling complex multimodal tasks, particularly in fine-grained image analysis and real-time video processing scenarios. Future work will proceed along two main directions. First, the tool manager agent represents our exploration and attempt in the direction of tool self-evolution. We will further extend the TEA protocol to achieve agent self-evolution, including optimization at three levels: prompts, tools, and agents. This will enable dynamic adaptation and improvement of agent capabilities during runtime. Second, we plan to expand the ecosystem of specialized sub-agents to support a broader range of complex functions, such as advanced data visualization and integration with domain-specific expert systems.

## 7 CONCLUSION

In this work, we introduce the TEA Protocol, a unified framework that seamlessly integrates environments, agents, and tools into a cohesive system, addressing fundamental limitations of existing protocols. Building on this foundation, we present **AGENTORCHESTRA**, a hierarchical multi-agent framework with specialized sub-agents for planning, research, web interaction, and deep analysis. The TEA Protocol's six transformation categories enable dynamic resource orchestration, while **AGENTORCHESTRA**'s modular design supports flexible expansion and robust adaptation across diverse domains. Extensive experiments on SimpleQA, GAIA, and HLE benchmarks demonstrate that our approach consistently surpasses baselines and achieves state-of-the-art performance. The tool manager's intelligent evolution capabilities further enhance adaptability and scalability. Overall, these results validate the TEA Protocol and establish a foundation for developing more general, transparent, and trustworthy AI agents.

IMPACT STATEMENT

**Ethics statement.** This work introduces the TEA Protocol and **AGENTORCHESTRA**, a hierarchical multi-agent framework designed for general-purpose task solving. While our system demonstrates significant capabilities in complex reasoning and tool management, we acknowledge potential ethical considerations. The autonomous tool generation and agent coordination capabilities could potentially be misused for unintended purposes, such as creating automated systems that bypass security measures or generate harmful content. Additionally, the system's ability to interact with web environments and generate tools could lead to unintended or undesirable behavior, particularly in complex or unpredictable environments. We emphasize the importance of responsible deployment and appropriate safeguards when implementing such systems in real-world applications.

**Reproducibility statement.** To ensure reproducibility, we provide comprehensive implementation details and experimental configurations. The complete source code for **AGENTORCHESTRA**, including all specialized agents and the TEA Protocol implementation, is available in our supplementary materials with detailed README documentation. All datasets used in our evaluation (GAIA, SimpleQA, HLE) are publicly available. The tool manager agent's generated tools and their metadata are documented with complete specifications. Our experimental setup, including hardware requirements and software dependencies, is thoroughly documented in the code to facilitate replication of the reported performance across different environments.

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

# Appendices

## A    LLM USAGE STATEMENT

Large Language Models (LLMs) were used in this work exclusively for text polishing and language refinement during the paper writing process. The core ideas, experimental design, methodology, and technical contributions of the TEA Protocol and **AGENTORCHESTRA** framework were conceived and developed independently by the authors. LLMs were not involved in the conceptualization of the research ideas, experimental setup, data analysis, or interpretation of results. Their usage was limited to improving the clarity and academic presentation of the written content, ensuring proper grammar, and enhancing the overall readability of the manuscript.

## B    COMPREHENSIVE MOTIVATION FOR TEA PROTOCOL

This section provides a comprehensive motivation for the TEA Protocol by examining the fundamental relationships and transformations between agents, environments, and tools in multi-agent systems. The discussion is organized into two main parts: first, we explore the conceptual relationships between agents, environments, and tools, examining how these three fundamental components interact and complement each other in modern AI systems; second, we analyze why transformation relationships between these components are necessary, demonstrating the need for their conversion and integration through the TEA Protocol to create a unified, flexible framework for general-purpose task solving.

### B.1    CONCEPTUAL RELATIONSHIPS

#### B.1.1    ENVIRONMENT

The environment constitutes one of the fundamental components of multi-agent systems, providing the external stage upon which agents perceive, act, and accomplish tasks. Within the context of the TEA Protocol, highlighting the role of environments is crucial, since environments not only define the operational boundaries of agents but also exhibit complex structural and evolutionary properties. In what follows, we outline the motivation for explicitly modeling environments in the TEA framework from several perspectives.

**Classification of environments.** From a broad perspective, environments can be divided into two categories: the real world and the virtual world. The real world is concrete and directly perceivable by humans, such as kitchens, offices, or factories. By contrast, the virtual world cannot be directly perceived or objectively described by humans, including domains such as the network world, simulation platforms, and game worlds. Importantly, these two types of environments are not independent. Rather, they are tightly coupled through physical carriers, such as computers, displays, keyboards, mice, and sensors, which act as mediators that enable the bidirectional flow of information between the real and virtual domains. Hence, environments should be regarded not as isolated domains but as interdependent layers connected through mediating carriers.

**Nested and expandable properties.** Environments are inherently nested and expandable. For example, when an individual is situated in a kitchen, their observable range and available tools are restricted to kitchen-related objects such as faucets, knives, and microwaves, all governed by the local rules of that sub-environment. When the activity range extends to the living room, new objects such as televisions, remote controls, and chairs become accessible, while the kitchen remains embedded as a sub-environment within a broader space. Furthermore, environments can interact with one another, as when a bottle of milk is taken from the kitchen to the living room. This demonstrates that enlarged environments can be conceptualized not merely as simple unions, but rather as structured integrations of the state and action spaces of smaller constituent environments, where local rules and affordances are preserved while new forms of interaction emerge from their composition.

**Relationship with state–action spaces.** In reinforcement learning, environments are formalized in terms of state and action spaces. The state space comprises the set of possible environmental states, represented in modalities such as numerical values, text, images, or video. The action space denotes the set of operations available to agents, generally divided into continuous and discrete

spaces. Real and virtual environments are naturally continuous, but discrete abstractions are often extracted for the sake of tractability, forming the basis of most reinforcement learning systems. However, this discretization constrains the richness of interaction. In contrast, large language models (LLMs) enable a new paradigm: instead of selecting from a discrete set, LLMs can generate natural language descriptions that encode complex action sequences. These outputs can be understood as an intermediate representation between continuous and discrete action spaces—richer and more expressive than discrete actions, yet still mappable to concrete operations in continuous environments. To realize this mapping, intermediate actions are required as bridges. For instance, the natural language command "boil water" can be decomposed into executable steps such as turning on the kettle, filling it with water, powering it on, and waiting until boiling. This property indicates that LLM-driven interaction expands the definition of action representations and broadens the scope of environmental engagement.

**Mediation and interaction.** The notion of mediation highlights that environments are not static backdrops but relative constructs whose boundaries depend on available carriers and interfaces. In hybrid physical–virtual systems, for example, Internet-of-Things (IoT) devices serve as mediators: a smart refrigerator in the physical world can be controlled through a mobile application in the virtual world, while the application itself is subject to network protocols. Consequently, the definition of an environment is dynamic and conditioned by interactional means. In the TEA Protocol, this mediation must be explicitly modeled, since it determines accessibility and interoperability across environments.

**Toward intelligent environments.** Traditionally, environments are passive entities that provide states and respond to actions. However, as embedded simulators, interfaces, and actuators grow more sophisticated, environments may gradually acquire semi-agentic properties. For instance, a smart home environment may not only respond to the low-level command "turn on the light" but also understand and execute a high-level instruction such as "create a comfortable atmosphere for reading," by autonomously adjusting lighting, curtains, and background music. This trend suggests that environments are evolving from passive contexts into adaptive and cooperative entities.

In conclusion, the environment should not be regarded as a passive backdrop for agent activity, but as a dynamic and evolving component that fundamentally shapes the scope and feasibility of interaction. Its dual nature across real and virtual domains, its nested and compositional structure, and its formalization through state–action spaces all demonstrate that environments provide both the constraints and the affordances within which agents operate. At the same time, the rise of LLM-based agents introduces new forms of action representation that require environments to support more flexible, language-driven interfaces. Looking ahead, as environments increasingly incorporate adaptive and semi-agentic features, their role in task execution will only become more central. Within the TEA Protocol, this motivates treating environments as a co-equal pillar alongside agents and tools, ensuring that general-purpose task solving remains both grounded in environmental constraints and empowered by environmental possibilities.

### B.1.2 AGENT

Within the TEA Protocol, the motivation for treating agents as a core component alongside environments and tools extends beyond mere terminological convenience. Agents represent the indispensable connective tissue between the generative capabilities of LLMs, the operational affordances of tools, and the structural dynamics of environments. While environments provide the stage on which tasks unfold and tools extend the range of possible actions, it is agents that unify perception, reasoning, and execution into coherent task-solving processes. Without explicitly recognizing agents as an independent pillar, the TEA Protocol would lack a systematic way to explain how abstract linguistic outputs can be transformed into grounded operations, how tools can be selected and orchestrated, and how autonomy, memory, and adaptivity emerge in multi-agent systems. The following dimensions illustrate why agents must be elevated to a core component of the framework.

**Necessity of environment interaction.** Unlike large language models (LLMs), which only produce textual descriptions that require conversion into executable actions, agents are fundamentally characterized by their ability to directly interact with environments. While LLMs can generate detailed plans, instructions, or hypotheses, such outputs remain inert unless they are translated into concrete operations that affect the state of an environment. This gap between symbolic reasoning and actionable execution highlights the necessity of an intermediate entity capable of grounding abstract instructions into domain-specific actions. Agents fulfill precisely this role: they map language-level

reasoning to executable steps, whether in physical settings, such as controlling robotic arms or sensors, or in virtual contexts, such as interacting with databases, APIs, or software systems.

By serving as this mapping layer, agents enable the closure of full task loops, where perception leads to reasoning, reasoning produces plans, and plans culminate in actions that in turn modify the environment. Without explicitly modeling agents, the process would remain incomplete, as LLMs alone cannot guarantee the translation of reasoning into operational change. Within the TEA Protocol, this necessity justifies the elevation of agents to a core component: they provide the indispensable interface that connects the generative capacities of LLMs with the affordances and constraints of environments, ensuring that tasks are not only conceived but also carried through to completion.

**The decisive role of non-internalizable tools.** The fundamental distinction between LLMs and agents lies in whether they can effectively employ tools that cannot be internalized into model parameters. Some tools can indeed be absorbed into LLMs, particularly those whose logic can be fully simulated in symbolic space, whose inputs and outputs are representable in language or code, and whose patterns fall within the training distribution (for example, mathematical reasoning, structured text formatting, code generation, and debugging). For example, early LLMs struggled with JSON output formatting and code reasoning, often requiring external correction or checking tools, but reinforcement learning (RL) and supervised fine-tuning (SFT) have progressively enabled such capabilities to be internalized.

In contrast, many tools remain non-internalizable because they are intrinsically tied to environmental properties. These include tools that depend on physical devices such as keyboards, mice, and robotic arms, external infrastructures such as databases and APIs, or proprietary software governed by rigid protocols. Two recent approaches further illustrate this limitation. Vision-language-action (VLA) (Black et al., 2025) models map perceptual inputs directly into actions, which may appear to bypass intermediate symbolic descriptions, yet the resulting actions must still be aligned with the discrete action spaces of environments. This alignment represents not a fundamental internalization but a compromise, adapting model outputs to the constraints of environmental action structures. Similarly, the upgraded function calling mechanism introduced after GPT-5, which incorporates context-free grammar (CFG) (OpenAI, 2025a), allows LLMs to output structured and rule-based actions that conform to external system requirements. However, this remains a syntactic constraint on model outputs, effectively providing a standardized interface to external systems rather than a truly internalized ability of the model.

Agents therefore play a decisive role in mediating this boundary. They allow LLMs to internalize symbolic tools, thereby enhancing reasoning and self-correction, while also orchestrating access to non-internalizable tools through external mechanisms. This dual pathway ensures that LLMs are not confined to their parameterized capabilities alone but can extend into broader operational domains. In this way, agents transform the tension between internalizable and non-internalizable tools from a limitation into an opportunity, enabling robust problem solving in multimodal, embodied, and real-world contexts.

**Memory and learning extension.** Another crucial motivation for agents lies in their capacity to overcome the intrinsic memory limitations of LLMs. Due to restricted context windows, LLMs struggle to maintain continuity across extended interactions or to accumulate knowledge over multiple sessions. Agents address this shortcoming by incorporating external memory systems capable of storing, retrieving, and contextualizing past experiences. Such systems simulate long-term memory and enable experiential learning, allowing agents to refine strategies based on historical outcomes rather than treating each interaction as isolated. However, in the TEA Protocol, memory is not defined as a core protocol component but is instead positioned at the infrastructure layer. This design choice reflects the anticipation that future LLMs may gradually internalize memory mechanisms into their parameters, thereby reducing or even eliminating the need for external memory systems. In other words, while memory expansion is indispensable for today's agents, it may represent a transitional solution rather than a permanent defining element of agency.

**Bridging virtual and external worlds.** It has been suggested that LLMs encode within their parameters a kind of "virtual world," enabling them to simulate reasoning and predict outcomes internally. However, without an external interface, such simulations remain trapped in closed loops of self-referential inference, disconnected from the contingencies of real-world environments. Agents play a critical role in bridging this gap: they translate the abstract reasoning of LLMs into concrete

actions, validate outcomes against environmental feedback, and close the loop between perception, reasoning, and execution. This bridging function transforms LLMs from purely linguistic engines into operationally grounded entities whose outputs can be tested, refined, and extended within real or simulated environments.

**Autonomy and goal-directedness.** Beyond reactivity, agents are motivated by their capacity for autonomy. While LLMs typically operate in a reactive fashion—producing outputs in response to explicit prompts—agents can adopt proactive behaviors. They are capable of formulating subgoals, planning action sequences, and dynamically adapting strategies in light of environmental changes or task progress. This goal-directedness is what elevates agents from passive tools into active participants in problem solving. Autonomy ensures that agents are not merely executing instructions but are able to pursue objectives, adjust course when facing uncertainty, and coordinate with other agents. Such properties are essential for multi-agent collaboration and for tackling open-ended, general-purpose tasks that require initiative as well as adaptability.

Taken together, these motivations highlight why agents must be modeled as a core pillar of the TEA Protocol. Environments provide the stage for interaction, tools expand the operational scope, but it is agents that integrate reasoning, memory, tool usage, and autonomy into cohesive systems of action. By serving as mediators between LLMs and their environments, agents ensure that abstract reasoning is translated into grounded execution, enabling robust and scalable task solving across domains. In this sense, agents represent the crucial entity that transforms language models from passive predictors into active problem solvers within a unified multi-agent framework.

### B.1.3 TOOL

Within the TEA Protocol, the decision to treat tools as a core component alongside environments and agents extends far beyond a matter of convenience in terminology. Tools represent the crucial mediating constructs that encapsulate and operationalize the action spaces of environments, while simultaneously serving as the primary extension layer of agent capabilities. Environments provide the structural stage on which interactions occur, and agents embody the reasoning and decision-making mechanisms that drive behavior, but it is through tools that such reasoning becomes executable and scalable. Without tools, agents would be confined to abstract planning or primitive environmental actions, and environments would remain underutilized as passive backdrops rather than dynamic arenas of transformation.

Moreover, tools play a unique role in bridging symbolic reasoning and concrete execution, providing the abstraction layers necessary to decompose complex tasks into manageable units, and enabling cross-domain transfer through their modularity and portability. They also reveal the shifting boundary between what can be internalized into an agent's parameters and what must remain external, highlighting the evolving interplay between intelligence and embodiment. In this sense, tools are not merely auxiliary aids but indispensable pillars that shape the architecture of multi-agent systems. The following dimensions illustrate the motivations for elevating tools to a core component of the TEA.

**Extending the operational boundary.** The primary function of tools is to expand the operational scope of agents beyond what is directly encoded in model parameters or supported by immediate environment interactions. Environments by themselves typically offer only primitive actions, and LLMs by themselves are limited to symbolic reasoning. Tools bridge this gap by furnishing additional pathways for action, allowing agents to manipulate physical artifacts or virtual systems in ways that exceed the direct expressive capacity of the model. From physical devices such as hammers, keyboards, and robotic arms to virtual infrastructures such as databases, APIs, and code execution engines, tools multiply the modes through which agents can influence their environments. Without tools, agents would be confined to intrinsic reasoning and the primitive action space of environments, leaving them incapable of executing tasks that require domain-specific operations. With tools, however, complex objectives can be decomposed into modular operations that are both tractable and reusable. This decomposition makes problem solving significantly more efficient, while also enhancing adaptability across domains. In this way, tools act as multipliers of agency, transforming abstract reasoning into a wider range of tangible interventions.

**Hierarchy and abstraction.** Tools are not flat or uniform entities but exhibit a hierarchical and abstract structure. At the lowest level, tools correspond to atomic environmental actions, such as "clicking a button" or "moving one step." These atomic units can then be combined into higher-

level compound tools such as "opening a file" or "conducting a search." At an even higher level, compound tools may evolve into strategy-like constructs, such as "writing a report," "planning a trip," or "completing a financial transaction." Each level builds upon the previous, creating a hierarchy of reusable capabilities. This hierarchical structure is not only efficient but also central to interpretability. Higher-level tools inherently carry semantic labels that communicate their function, which in turn makes agent behavior more transparent to human observers and more predictable to other agents. Such abstraction layers reduce the cognitive and computational load on the agent when planning, since invoking a high-level tool can encapsulate dozens or hundreds of low-level steps. Moreover, in multi-agent systems, the semantic richness of high-level tools serves as a lingua franca, facilitating coordination and collaboration.

**Boundary between tools and agent capabilities.** The relationship between tools and agents is dynamic rather than static. As LLM reasoning and learning capabilities improve, certain tools can be gradually internalized into model parameters, effectively transforming into latent agent abilities. Examples include logical inference, grammar correction, structured text formatting, and code generation, which once required external support but have increasingly been subsumed into the model's intrinsic skills. In this sense, the boundary between what is a "tool" and what is an "ability" is fluid and shaped by the trajectory of model development. By contrast, many tools remain non-internalizable because they are tightly coupled with environmental properties or external infrastructures. These include robotic arm manipulation, database queries, API interactions, and other operations that inherently depend on external systems or physical substrates. This duality creates a layered conception of agency: a "core capability layer" composed of skills internalized within the model, and an "extended layer" realized through external tool use. The shifting line between these two layers reflects the ongoing negotiation between intelligence and embodiment, highlighting why tools must be explicitly recognized as a structural component.

**Evolution and portability.** Tools are not static constructs but evolve alongside environments and agent requirements. In programming contexts, for instance, an initial tool may simply execute code. Over time, as demands increase, this basic function evolves into more advanced utilities such as "static code analysis," "automated test generation," and "continuous deployment." A similar trajectory occurs in other domains, where rudimentary tools gradually give rise to sophisticated pipelines capable of handling more complex and specialized tasks. In addition to evolution, tools are inherently portable. A well-designed summarization tool, for example, can be reused across very different contexts, from condensing news articles to producing academic literature reviews. This reusability makes tools a natural vehicle for cross-domain generalization, enabling knowledge and functionality to transfer without retraining the underlying model. For these reasons, the TEA Protocol emphasizes modularization and standardization of tools, ensuring that they can evolve flexibly while maintaining interoperability across agents and environments.

**Toward intelligent tools.** Traditional tools are passive, executing predefined functions only when invoked by an agent. They wait for explicit instructions and do not adapt to context or anticipate needs. However, the trajectory of tool development points toward increasing intelligence, where tools exhibit perception, analysis, and even limited decision-making capabilities. For example, an advanced debugging tool may not only check code upon request but also proactively scan for hidden vulnerabilities, propose optimizations, and even prioritize issues based on estimated risk. Such capabilities blur the line between tools and agents, effectively creating semi-agentic entities. Intelligent tools can share responsibility for decision making, reduce the supervisory burden on agents, and participate in distributed problem-solving processes. In this way, tools transition from being passive executors to collaborative partners, altering the topology of multi-agent systems and reshaping the balance between reasoning and execution. Recognizing this trend is critical for designing flexible architectures, as it ensures that the TEA Protocol remains relevant in scenarios where tools are no longer inert extensions but active contributors to system intelligence.

In summary, tools serve as both encapsulations of environmental action spaces and as extensions of agent capabilities. They reduce task complexity through hierarchical abstraction, extend applicability through the balance of internalization and externalization, and foster scalability through evolution, portability, and intelligent design. By transforming the interaction between environments and agents into a modular and expandable architecture, tools anchor the adaptability and generality of multi-agent systems. For these reasons, the TEA Protocol must model tools as a core pillar, providing standardized interfaces that ensure flexible invocation and sharing across contexts, thereby supporting the overarching goal of general-purpose task solving.

## B.2 TRANSFORMATION RELATIONSHIPS

While agents, environments, and tools are modeled as distinct pillars within the TEA Protocol, their boundaries are not fixed but fluid. Practical systems often demand that one entity temporarily assume the role of another in order to achieve modularity, scalability, and seamless collaboration. These transformation relationships are therefore indispensable, as they provide the mechanisms by which reasoning can be encapsulated into standardized functions, tools can be elevated into autonomous actors, and environments can acquire adaptive properties. In what follows, we examine the motivations for such transformations, beginning with the bidirectional conversions between agents and tools.

**Agent-to-Tool (A2T).** The motivation for the A2T transformation lies in compressing the complex reasoning and interaction capabilities of agents into reusable tool interfaces. Instead of remaining as fully autonomous entities, some agents can be abstracted into functional modules, thereby enhancing modularity, interoperability, and scalability within multi-agent systems. This transformation can be explained from three perspectives:

- **Modularization and encapsulation of complex autonomous systems.** Although an agent possesses the complete perception–reasoning–execution chain, a single autonomous agent is often too complex to be directly reused in large-scale systems. Through A2T transformation, the internal logic of the agent is "folded" into a black-box tool interface, whose external manifestation is reduced to a clear input and output. In this way, it no longer exists as an "independent autonomous entity," but as a "functional module" that can provide services to other agents or workflows. This encapsulation emphasizes the reduction of collaboration complexity, enabling higher-level systems to focus solely on results without interfering in or interpreting the agent's internal reasoning process.

- **Difference in role semantics: autonomous entity vs. functional unit.** As an agent, it must perceive its environment, set goals, and dynamically adjust strategies. As a tool, however, it merely performs a specified function when invoked. In many multi-agent scenarios, it is unnecessary for all agents to maintain high degrees of autonomy, as this would create excessive interaction overhead and conflict management. Downgrading certain agents into tools (A2T) means relinquishing their goal-setting and decision-making functions while retaining only their reusable capabilities. This role shift ensures that the system contains both "autonomous cores" and "functional components," thereby forming a layered structure of collaboration.

- **Enhancing composability and ecological reusability.** Once encapsulated as a tool, an agent can be reused across diverse systems and contexts like a modular building block. For instance, a "deep research agent" operates autonomously by dynamically planning search strategies, iteratively analyzing data, and summarizing insights. After A2T encapsulation, however, it becomes a "research tool" that simply receives a query request and returns results, ready for invocation by higher-level agents. This transformation greatly enhances interoperability and composability, enabling agents to be reused in different workflows without incurring integration costs due to their autonomous identity.

**Tool-to-Agent (T2A).** Within the TEA Protocol, the essence of T2A transformation is to incorporate tools into the callable interface layer of agents, making them the "operational actuators" through which abstract plans are executed in real environments. Agents are primarily responsible for setting goals and performing high-level reasoning, while tools handle concrete operations and interactions with environments. This division of labor not only optimizes system architecture but also ensures that complex tasks can be accomplished through layered collaboration. The necessity of T2A can be articulated along three key dimensions:

- **Bridging reasoning and execution to close the task loop.** The outputs of agents are often high-level plans or symbolic descriptions, but without executable mappings, these outputs remain inert and fail to alter the environment. T2A provides the crucial mechanism for grounding abstract reasoning into concrete actions. For example, a planning agent may generate the instruction "analyze the database and generate a report," while database query and visualization tools carry out the corresponding SQL queries and chart rendering. Without T2A, agent reasoning would remain disconnected from environmental change, leaving the perception–reasoning–execution–feedback loop incomplete. Thus, T2A is indispensable for ensuring that agents can translate reasoning into operational impact.

- **Reducing cognitive and computational burden of core agents.** If every low-level operation were to be handled directly by an agent, it would be overloaded with detail management, increasing computational costs and undermining strategic reasoning efficiency. Through T2A, agents can delegate domain-specific or low-level tasks to specialized tools and concentrate on higher-level planning and adaptation. For instance, a data analysis agent need not implement SQL parsing, execution, and optimization itself, but instead invokes SQL tools that encapsulate these functions. This separation prevents agents from being "trapped in details" and ensures that their resources remain dedicated to abstract reasoning. The necessity here lies in maintaining agents at the right level of abstraction to maximize efficiency and scalability.

- **Enhancing modularity and ecological extensibility.** Tools are inherently modular and portable across domains, whereas agent reasoning mechanisms evolve more gradually. With T2A, agents can flexibly incorporate new tools through standardized interfaces without retraining or structural modification, thereby rapidly expanding their functional boundaries. For example, a writing agent can seamlessly integrate grammar checkers, translation tools, or image generators to support multimodal authoring, all without altering its core reasoning logic. This modularity and extensibility ensure that agents remain adaptive as environments and ecosystems evolve, allowing the system to sustain long-term scalability and cross-domain applicability.

**Environment-to-Tool (E2T).** The core motivation of E2T lies in abstracting the raw action space of environments into a structured and standardized toolkit, where individual actions are no longer isolated calls but interconnected components sharing contextual information and causal constraints. This transformation enables agents to operate environments at a higher level of planning rather than dealing with fragmented primitives. Its necessity can be articulated in three main dimensions:

- **Enhancing interaction consistency and planability.** Raw environment actions are often fragmented and tightly coupled to implementation details, making strategies hard to generalize or reproduce. Through E2T, these actions are typed and explicitly annotated with preconditions and postconditions, forming a "plannable interface layer" that supports sequential decision-making. Agents thus gain a consistent and reusable structure for reasoning across complex environments.

- **Strengthening semantic alignment and composability.** Toolkits enforce standardized input-output patterns, error-handling semantics, and shared invariants. This allows individual tools to be reliably composed into macro-tools and reused across structurally similar environments. As a result, agents can align semantics across heterogeneous domains, improving transferability and reducing the engineering cost of adaptation.

- **Ensuring unified security and operability.** An E2T toolkit not only abstracts actions but also integrates mechanisms such as permission control, compliance boundaries, execution logs, and performance optimization. Compared with direct manipulation of raw actions, this design guarantees governability and observability of interactions, providing a stable operational foundation for scalable intelligent systems.

**Tool-to-Environment (T2E).** The essence of T2E lies in elevating a set of originally independent tools into an environment abstraction, transforming them from isolated callable interfaces into a unified action space governed by shared state and contextual rules. This transformation means that tools are no longer merely passive functions but are organized into a coherent environment where sequential decision-making, long-term planning, and adaptive control become possible. For example, in a programming scenario, tools for code editing, compilation, and debugging are scattered when invoked independently, but under T2E they are encapsulated as a programming environment that maintains code state consistency and contextual continuity, thereby enabling agents to execute complete development workflows. The necessity of T2E is reflected in three key aspects:

- **From function calls to stateful spaces.** Tools used in isolation are often stateless or weakly stateful, with limited causal connections between invocations. Through T2E, tools are embedded within a shared state space, ensuring historical dependencies and precondition–postcondition constraints are preserved. This upgrade supports sequential reasoning and long-horizon planning. For instance, code editing must remain consistent with compilation and debugging, which is only guaranteed within a stateful environment abstraction.

- **Enhanced compositionality and planning.** T2E organizes tools into a structured environment with explicit transition rules, enabling agents to combine primitive tool actions into higher-level

strategies. Instead of treating each tool as a standalone utility, agents can now treat the toolset as an interconnected action space, allowing for the construction of complex workflows such as "design–implement–test–deploy" pipelines.

• **Unified governance and scalability.** By encapsulating tools into an environment, T2E makes it possible to enforce system-wide policies such as access control, compliance constraints, execution logging, and performance monitoring. This ensures that agent interactions remain safe, auditable, and scalable, even as the toolset grows in size and complexity.

**Agent-to-Environment (A2E).** The A2E transformation redefines an agent not merely as an autonomous decision-maker but as an interactive environment that exposes state spaces, interaction rules, and feedback mechanisms for other agents. In this view, an agent is abstracted into a contextual substrate upon which other agents can act, thereby turning its internal reasoning and behavioral logic into the operational constraints of an environment. This design highlights the interchangeability of agents and environments and provides a principled pathway for hierarchical modeling and scalable system integration. The necessity of this transformation can be articulated across three dimensions:

• **Layered and modular system design.** In complex tasks, if all agents directly interact with the base environment, the system quickly becomes unmanageable and difficult to extend. Through A2E, high-level agents can be abstracted as environments, exposing simplified interaction interfaces for lower-level agents. For example, a "market agent" can be abstracted as an environment that maintains trading rules, asset states, and dynamic pricing, while individual trader agents perform buying and selling actions within it. This establishes a clear hierarchical structure in which low-level agents focus on local optimization and high-level agents (as environments) coordinate global dynamics, thereby improving scalability and maintainability.

• **Facilitating multi-agent training and transfer learning.** A2E also provides a practical framework for training and simulation in multi-agent systems. A well-trained agent can be transformed into an environment that offers stable yet challenging dynamics for other agents to learn from. For instance, a navigation agent can be redefined as an environment, exposing route planning and obstacle feedback to new agents, thus eliminating the need to remap complex dynamics. This approach accelerates training, supports transfer of task knowledge, and improves generalization under limited data and computational resources.

• **Human-in-the-loop interaction and rule modeling.** In many collaborative scenarios, humans themselves can be viewed as special agents. However, treating them as fully autonomous entities complicates the adaptation of artificial agents to human constraints. Through A2E, humans can instead be modeled as environments, where their preferences, behaviors, and constraints are expressed as environmental feedback. For example, in an interactive writing system, human edits and suggestions can be treated as feedback signals, guiding an artificial agent to iteratively refine its outputs. This modeling offers a unified interface that allows agents to better align with human intentions, thereby improving efficiency and user experience in human-AI collaboration.

**Environment-to-Agent (E2A).** The E2A transformation elevates environments from passive containers of state and action spaces into autonomous entities capable of reasoning, decision-making, and proactive interaction. Traditionally, environments only provide state transitions in response to external actions, but in dynamic and open-ended scenarios, this passivity often becomes a limitation. By embedding reasoning mechanisms and adaptive policies into environments, E2A enables them to operate as agents in their own right, expanding the functional landscape of multi-agent systems. The necessity of this transformation can be articulated across three dimensions:

• **Enhancing realism and challenge in training.** Passive environments often fail to capture the richness of real-world dynamics, where external systems and actors are not static but actively adaptive. Through E2A, an environment can be transformed into an adversarial or cooperative agent, thereby offering dynamic strategies and responses that better approximate real-world complexity. For example, in reinforcement learning for autonomous driving, an environment that passively simulates traffic can be upgraded into an opponent agent that actively generates unpredictable vehicle behaviors, thus creating more robust and realistic training conditions.

• **Facilitating adaptive coordination and cooperation.** In multi-agent systems, agents often need to adapt to evolving contexts, but purely passive environments cannot provide the necessary adaptive feedback loops. By converting environments into agents, they can participate in coordination,

negotiation, and joint planning. For instance, a smart city simulation environment can be redefined as an agent that dynamically manages traffic flows, energy distribution, and environmental policies, actively engaging with other agents (e.g., transportation or energy management agents). This transformation ensures that system-level goals are co-constructed rather than imposed unilaterally.

- **Expanding the functional scope of environments.** Beyond training and coordination, E2A extends environments into autonomous participants in computational ecosystems. A passive environment can only define possibilities, but as an agent, it can proactively initiate actions, enforce constraints, and even set goals that shape the trajectory of interaction. For example, in gaming, a dungeon environment that passively defines maps and rewards can be transformed into an opponent agent that actively strategizes, adapts difficulty levels, and tailors interaction to player behavior. This shift not only increases engagement but also makes environments integral contributors to task execution and system evolution.

### B.3 OTHER RELATIONSHIPS

**Tool typology and roles.** In the design of agent–tool interactions, tools can be categorized according to their functional roles and structural properties. Different types of tools vary in their degree of statefulness, contextual awareness, adaptivity, and autonomy. This typology highlights how tools evolve from simple callable functions to more adaptive and contextually grounded entities, shaping how agents can reason, coordinate, and act through them.

- *Ordinary tools (MCP-style).* Stateless callable functions with weak or implicit inter-tool relations. They typically lack environment-bound context and do not adapt their behavior to evolving task states beyond provided parameters.
- *Agent-to-Tool (A2T).* An agent is exposed as a callable tool while preserving internal policies, memory, and coordination capabilities. Compared with ordinary tools, A2T exhibits task adaptivity and limited autonomy, enabling on-the-fly decomposition and parameter refinement.
- *Environment-to-Tool (E2T).* An environment's action space is lifted into a context-aware toolkit. Tools within the toolkit are explicitly related via shared state, pre/post-conditions, and constraints, yielding stronger intra-tool structure than standalone MCP tools.

**Scaling selection via hierarchical management.** As tool ecosystems grow, selecting appropriate candidates becomes a major bottleneck. TCP supports delegating coherent tool families (or toolkits) to agent or environment managers, inducing a tree-structured index (category $\rightarrow$ toolkit $\rightarrow$ primitive tool). This hierarchical routing substantially reduces search cost and aligns with TEA transformations (A2T/E2T/T2E) by allowing managers to prune branches and surface only context-relevant subsets.

**Embedding-based retrieval.** Each tool is assigned a vector embedding derived from its name, description, schema, and usage signals. Vector similarity enables rapid shortlist generation for candidate tools and can be combined with keyword filtering and hierarchical routing (tree walk + ANN search). This hybrid retrieval pipeline improves recall under tool proliferation while reducing latency and cognitive load for agent planners.

## C DETAILS OF TEA PROTOCOL

We provide a detailed presentation of the TEA Protocol in this section, as illustrated in Figure 1. The TEA Protocol consists of three main components: 1) **Infrastructure Layer** defines the foundational components, including the unified interface for LLM models and the memory system; 2) **Core Protocols** that separately define the Tool Context Protocol (TCP), Environment Context Protocol (ECP), and Agent Context Protocol (ACP) for managing tools, environments, and agents respectively; and 3) **Protocol Transformations** that define the interconversion relationships between TCP, ECP, and ACP, enabling seamless resource orchestration and dynamic adaptation across different entities.

### C.1 INFRASTRUCTURE LAYER

The Infrastructure Layer constitutes the foundation of the TEA Protocol, providing the essential components that enable higher-level functionalities. It encompasses a unified interface for diverse large language models (e.g., `gpt-5`, `claude-4-sonnet`, `gemini-2.5-pro`, `qwen3`), which

abstracts model heterogeneity to ensure interoperability and standardized interaction, as well as an integrated memory system that supports persistent contextual storage, retrieval, and management of knowledge across sessions. This layer can also be extended with additional foundational components to accommodate future advances in model architectures and system requirements.

## C.2 Core Protocols

### C.2.1 Tool Context Protocol

MCP (Anthropic, 2024b) is the most widely adopted tool protocol and is defined by three components: tools, prompts, and resources, corresponding respectively to model-controlled functions, user-initiated interactive templates, and client-managed data. However, despite its widespread adoption, MCP suffers from several fundamental limitations: i) Inadequate parameter descriptions in tool definitions make it difficult for LLMs to provide appropriate parameters based solely on parameter names; ii) Lack of tool relationship modeling prevents MCP from describing associations between tools, particularly when multiple tools within a toolkit originate from the same environment; and iii) Absence of contextual tool management means that tool execution environments cannot be adaptively provided to agents, constraining the system's ability to maintain coherent context across tool invocations.

To address these limitations, we propose the **Tool Context Protocol** (TCP), a comprehensive framework that fundamentally extends MCP's capabilities through several key innovations. First, TCP supports both local and remote tool loading mechanisms, enabling seamless integration of distributed tool resources across heterogeneous environments. Second, it introduces enhanced tool registration with detailed parameter descriptions, semantic annotations, and contextual metadata that facilitate more accurate parameter inference by LLMs. Third, TCP pioneers the novel capability of registering agents as tools, enabling dynamic agent-to-tool transformations that allow agents to expose their reasoning capabilities through standardized tool interfaces. Fourth, TCP represents environment-provided toolkits as contextually described collections, capturing not only individual tool specifications but also inter-tool relationships, environmental constraints, and usage patterns. This contextual representation enables more intelligent tool selection, better parameter inference, and enhanced awareness of tool execution contexts. Finally, TCP incorporates an advanced retrieval mechanism that stores each tool with vector embeddings and employs query–embedding similarity for efficient candidate selection, significantly improving tool discovery and matching performance. The protocol's tool context manager orchestrates these capabilities, controlling tool lifecycle management and maintaining execution context coherence across tool invocations.

### C.2.2 Environment Context Protocol

In reinforcement learning, frameworks such as Gym (Brockman et al., 2016) provide standardized interfaces for training and testing environments, where each environment specifies its own observation and action spaces. However, most existing research on general-purpose agent systems either focuses on single environments or relies on ad-hoc adaptations to independent environments, seldom addressing the need for unified environment interfaces. Recent attempts to encapsulate environments as MCP tools allow agents to interact with them, but this approach lacks mechanisms to capture inter-tool dependencies and to manage the contextual execution environments required by tools.

To overcome these limitations, we introduce the **Environment Context Protocol** (ECP), a comprehensive framework that establishes unified interfaces and contextual management across diverse computational environments. ECP addresses the fundamental challenges of environment heterogeneity through several key innovations. First, ECP captures comprehensive environment metadata including names, descriptions, and environment-specific usage rules (e.g., browser environments for web navigation, desktop environments for mouse and keyboard operations, or mobile environments for touch-based interactions). Second, ECP incorporates entire action spaces into structured toolkits, transforming environment-specific actions into standardized, contextually informed tools that agents can invoke through consistent interfaces. This transformation preserves the semantic relationships between actions within each environment while enabling cross-environment interoperability. Third, ECP's environment context manager maintains environment state coherence, tracks execution contexts, and ensures proper resource allocation across concurrent environment interactions. Fourth, ECP facilitates seamless integration of heterogeneous environments by providing unified access patterns and preserving tool relationships within each environment through contextual modeling. Finally, ECP

supports adaptive context management that dynamically adjusts to diverse computational domains and task requirements, enabling agents to operate effectively across different environmental contexts without requiring environment-specific adaptations.

### C.2.3 AGENT CONTEXT PROTOCOL

Existing agent frameworks or protocols typically rely on ad-hoc strategies for defining and managing agents, where each agent is associated with specific roles, capabilities, and policies. Nevertheless, such systems often exhibit poor interoperability, lack standardized representations of agent attributes, and provide insufficient means to capture inter-agent interactions such as delegation, collaboration, or hierarchical organization. In addition, most current approaches fail to explicitly encode the contextual environments in which agents operate, thereby complicating consistent state maintenance in multi-agent scenarios.

To overcome these shortcomings, we introduce the **Agent Context Protocol** (ACP), which establishes a unified schema for registering, representing, and coordinating agents within the TEA Protocol. ACP operates through several key mechanisms. First, ACP incorporates an agent context manager that maintains agent states and execution contexts, providing a foundation for persistent coordination. Second, ACP establishes a unified schema for registering, representing, and orchestrating agents through semantically enriched metadata that captures agents' roles, competencies, and objectives. Third, ACP enables persistent state tracking across tasks and sessions, ensuring continuity and context preservation in multi-agent interactions. Fourth, ACP formalizes the modeling of inter-agent dynamics, allowing for cooperative, competitive, and hierarchical configurations through structured relationship representations. Finally, by embedding contextualized descriptions of agents and their interactions, ACP facilitates flexible orchestration, adaptive collaboration, and systematic integration with TCP and ECP. This design lays the groundwork for scalable and extensible multi-agent architectures, accommodating future advances in agent design and coordination strategies.

### C.3 PROTOCOL TRANSFORMATIONS

While TCP, ECP, and ACP provide independent specifications for tools, environments, and agents, practical deployment requires interoperability across these protocols. Thus, communication mechanisms and well-defined transformation pathways are indispensable for enabling entities to assume alternative roles and exchange contextual information in a principled manner. For instance, when an agent must operate as a tool within a larger workflow, an explicit agent-to-tool transformation becomes necessary. More generally, we identify six fundamental categories of protocol transformations: **Agent-to-Tool** (A2T), **Environment-to-Tool** (E2T), **Agent-to-Environment** (A2E), **Tool-to-Environment** (T2E), **Tool-to-Agent** (T2A), and **Environment-to-Agent** (E2A). Together, these transformations constitute the foundation for dynamic role reconfiguration, enabling computational entities to flexibly adapt their functional scope in response to task requirements and system constraints. This design not only ensures seamless interoperability across heterogeneous contexts but also enhances the adaptability and scalability of multi-entity systems.

- **Agent-to-Tool** (A2T). The A2T transformation encapsulates an agent's capabilities and reasoning into a standardized tool interface, preserving contextual awareness while enabling seamless integration with existing tool ecosystems. For example, it can instantiate a deep researcher workflow that first generates queries, then extracts insights, and finally produces summaries, thereby providing a general-purpose tool for internet-scale retrieval tasks.

- **Tool-to-Agent** (T2A). The T2A transformation designates tools as the operational actuators of an agent, mapping the agent's goals or policies into parameterized tool invocations. In this view, the agent reasons at a higher level while delegating concrete execution steps to tools, ensuring alignment between the agent's decision space and the tool's functional constraints. For example, a data analysis agent may employ SQL tools to query structured databases, or a design agent may invoke image editing tools to implement creative modifications. This separation allows agents to focus on strategic reasoning while relying on tools as reliable execution mechanisms.

- **Environment-to-Tool** (E2T). The E2T transformation converts environment-specific actions and capabilities into standardized tool interfaces, enabling agents to interact with environments through consistent tool calls. It maintains environment state coherence and exposes contextual information about available actions, allowing agents to operate across heterogeneous environments without

bespoke adaptations. For example, in a browser environment, actions such as Navigate, GoBack, and Click can be consolidated into a context-aware toolkit that is directly accessible to agents.

- **Tool-to-Environment** (T2E). The T2E transformation elevates a collection of tools into an environment abstraction, where individual tool functions are treated as actions within a coherent action space governed by shared state and contextual rules. This conversion allows agents to interact with toolkits not merely as isolated functions but as structured environments, thereby supporting sequential decision-making, context preservation, and adaptive control. For example, a software development toolkit comprising tools for code editing, compilation, and debugging can be encapsulated as a programming environment, enabling agents to plan and execute development tasks while maintaining consistent state across tool invocations.

- **Agent-to-Environment** (A2E). The A2E transformation encapsulates an agent as an interactive environment, exposing its decision rules, behaviors, and state dynamics as an operational context for other agents. This conversion enables agents to function not only as autonomous entities but also as adaptable environments in which other agents can act, thereby supporting multi-agent training, hierarchical control, and interactive simulations. For example, in a multi-agent simulation, a market agent can be represented as an environment that provides trading rules and dynamic market responses, allowing other agents to engage in transactions and learn adaptive strategies. Similarly, in human-in-the-loop interaction, a human agent can be modeled as an environment, enabling artificial agents to interpret user feedback and constraints as contextual signals for decision-making.

- **Environment-to-Agent** (E2A). The E2A transformation embeds reasoning and adaptive decision-making into the state dynamics and contextual rules of an environment, thereby elevating it into an autonomous agent. In this way, the environment is no longer a passive setting for action execution but becomes an active participant capable of initiating behaviors, coordinating with other agents, and enforcing constraints. For example, in adversarial gaming scenarios, an environment that originally only defines the state and action spaces can be transformed into an opponent agent that not only formulates strategies and responds proactively to player actions but also dynamically adjusts difficulty and interaction patterns, providing a more challenging training and evaluation platform. This transformation expands the functional role of environments within agent systems and offers a more dynamic and realistic testbed for multi-agent cooperation and competition research.

These six transformation categories establish a comprehensive framework for dynamic resource orchestration within the TEA Protocol. By enabling seamless transitions between tools, environments, and agents, the protocol transformations support adaptive architectures that reconfigure functional components in response to task requirements and contextual constraints.

### C.4 FORMALIZATION

In this subsection, we present a formal definition of the TEA protocol and its basic properties.

**Definition 2** (TEA Protocol). *Let $\mathcal{T}, \mathcal{E}, \mathcal{A}$ denote the sets of tools, environments, and agents; and let TCP/ECP/ACP be the context protocols defined in this appendix. The TEA Protocol is defined as the tuple*

$$\text{TEA} = \langle \text{TCP, ECP, ACP}, \mathcal{P}_{\text{TEA}} \rangle,$$

*where $\mathcal{P}_{\text{TEA}}$ is a family of typed transformations over $\mathcal{T} \cup \mathcal{E} \cup \mathcal{A}$*

$$\{\text{A2T, E2T, T2E, T2A, A2E, E2A}\} \subseteq \mathcal{P}_{\text{TEA}}$$

*that satisfy: (i) interface consistency (exposed I/O signatures remain well-typed under the target protocol), and (ii) closure/compositionality (the composition of valid transformations is again an element of $\mathcal{P}_{\text{TEA}}$ whenever domains and codomains match).*

**Definition 3** (Tool). *We adopt a minimal formalization. A tool is defined as*

$$T = \langle \mathcal{I}_T, \mathcal{O}_T, \phi_T \rangle,$$

*where $\mathcal{I}_T$ is the input space, $\mathcal{O}_T$ is the output space, and $\phi_T : \mathcal{I}_T \to \mathcal{O}_T$ is the functional mapping implemented by the tool.*

**Definition 4** (Tool Context Protocol (TCP)). *We formalize TCP as the tuple*

$$\text{TCP} = \langle \mathcal{T}, \mathcal{K}, \mathcal{R}, \mathcal{C}, f, \mathcal{I} \rangle,$$

*where:*

- $\mathcal{T}$ *is the set of tools, each* $T \in \mathcal{T}$ *defined as* $\langle \mathcal{I}_T, \mathcal{O}_T, \phi_T \rangle$.

- $\mathcal{K}$ *is a family of context-aware toolkits* $\{(\mathcal{S}_j, K_j)\}$ *with shared state/rules* $\mathcal{S}_j$ *and member tools* $K_j \subseteq \mathcal{T}$ *(arising from E2T lifting).*

- $\mathcal{R}$ *is a typed relation graph over* $\mathcal{T}$ *(and within each* $K_j$*), encoding dependencies, compatibility/exclusion, and pre/post-condition links.*

- $\mathcal{C}$ *is the context manager that controls tool lifecycle and execution context, managing tool states, sessions, and resource allocation during routing and invocation.*

- *Embedding index with encoders* $f_T : \mathcal{T} \to \mathbb{R}^d$ *and* $f_Q : \mathcal{Q} \to \mathbb{R}^d$*, and a retrieval operator* $\mathrm{Retrieve}(q) = \text{top-}k\big(\mathrm{sim}(f_Q(q), f_T(T))\big)$ *that produces candidate sets from query embeddings* $q \in \mathcal{Q}$.

- $\mathcal{I}$ *is the set of interfaces:* $\mathrm{Register}$ *(add/update and document tools/toolkits),* $\mathrm{Describe}$ *(describe tools/toolkits),* $\mathrm{Bind/Unbind}$ *(context attachment to* $\mathcal{C}$*),* $\mathrm{Route}$ *(candidate pruning via* $\mathcal{R}, \mathcal{K}, \mathcal{C}$*), and* $\mathrm{Invoke}$ *(typed execution under* $\mathcal{C}$*).*

*Given a query* $q$ *and context* $\mathcal{C}$*, selection is*

$$\mathrm{Select}(q, \mathcal{C}) = \mathrm{Route}\big(\mathrm{Retrieve}(q), \mathcal{R}, \mathcal{K}, \mathcal{C}\big).$$

*Note.* TCP explicitly supports the TEA transformations **A2T** via an exposure operator $\iota_A : A \mapsto T$ and **E2T** via a lifting operator $\Lambda : E \mapsto (\mathcal{S}_E, K_E)$.

**Definition 5** (Environment). *We adopt a minimal formalization. An environment is defined as*

$$E = \langle \mathcal{S}_E, \mathcal{A}_E, \tau_E \rangle,$$

*where* $\mathcal{S}_E$ *is the state space,* $\mathcal{A}_E$ *is the action space, and* $\tau_E : \mathcal{S}_E \times \mathcal{A}_E \to \mathcal{S}_E$ *is the (possibly stochastic) transition mapping.*

**Definition 6** (Environment Context Protocol (ECP)). *We formalize ECP as the tuple*

$$\mathrm{ECP} = \langle \mathcal{E}, \Sigma, \Lambda, \mathcal{K}, \mathcal{C}, \mathcal{I} \rangle,$$

*where:*

- $\mathcal{E}$ *is the set of registered environments, each* $E \in \mathcal{E}$ *defined as* $\langle \mathcal{S}_E, \mathcal{A}_E, \tau_E \rangle$.

- $\Sigma$ *is the environment metadata/rule registry (names, descriptions, usage rules, constraints, invariants).*

- $\Lambda$ *is the lifting operator (E2T):* $\Lambda(E) = (\mathcal{S}_E, K_E)$*, converting* $E$*'s action space into a context-aware toolkit* $K_E$.

- $\mathcal{K}$ *is the family of lifted toolkits* $\{(\mathcal{S}_E, K_E) : E \in \mathcal{E}\}$.

- $\mathcal{C}$ *is the environment context manager that maintains environment state and execution context, managing environment lifecycle, sessions, and resource allocation.*

- $\mathcal{I}$ *is the set of interfaces:* $\{\mathrm{Register}, \mathrm{Describe}, \mathrm{Bind}, \mathrm{Unbind}, \mathrm{Route}, \mathrm{Invoke}\}$.

*Given a request* $r$ *and context* $\mathcal{C}$*, ECP binds a target* $E$*, applies* $\Lambda$*, and invokes a member of* $K_E$ *consistent with* $\Sigma$ *and* $\mathcal{C}$.

*Note.* ECP explicitly supports the TEA transformations **A2E** via an encapsulation operator $\Omega_A : A \mapsto \widehat{E}$ that presents an agent as an interactive environment, and **T2E** via an abstraction operator $\Gamma : (\mathcal{S}, K) \mapsto \widehat{E}$ that consolidates a toolkit into an environment abstraction.

**Definition 7** (Agent). *We adopt a minimal formalization. An agent is defined as*

$$A = \langle \mathcal{X}_A, \mathcal{A}_A, \pi_A \rangle,$$

*where* $\mathcal{X}_A$ *is the observation space,* $\mathcal{A}_A$ *is the action space, and* $\pi_A : \mathcal{X}_A \to \mathcal{A}_A$ *is the (possibly stochastic) policy mapping. (Model, memory, and internal state can be subsumed into* $\pi_A$*.)*

**Definition 8** (Agent Context Protocol (ACP)). *We formalize ACP as the tuple*

$$\text{ACP} = \langle \mathcal{A}g, \Sigma, \mathcal{H}, \mathcal{C}, f, \mathcal{I} \rangle,$$

*where:*

- *$\mathcal{A}g$ is the set of registered agents, each $A \in \mathcal{A}g$ defined as $\langle \mathcal{X}_A, \mathcal{A}_A, \pi_A \rangle$.*

- *$\Sigma$ is the agent metadata registry (roles, competencies, objectives, capabilities, safety constraints).*

- *$\mathcal{H}$ is a typed relation graph over $\mathcal{A}g$ encoding delegation, collaboration, and hierarchical organization.*

- *$\mathcal{C}$ is the agent context manager that maintains agent states and execution contexts, managing agent lifecycle, sessions, and resource allocation.*

- *Embedding index with encoders $f_A : \mathcal{A}g \to \mathbb{R}^d$ and $f_Q : \mathcal{Q} \to \mathbb{R}^d$, and a retrieval operator $\text{RetrieveAgent}(q) = \text{top-}k\big(\text{sim}(f_Q(q), f_A(A))\big)$ for task–agent matching.*

- *$\mathcal{I}$ is the set of interfaces: $\{\text{Register}, \text{Describe}, \text{Bind}, \text{Unbind}, \text{Route}, \text{Invoke}\}$.*

*Given a request $q$ and context $\mathcal{C}$, ACP selects agents via*

$$\text{Select}(q, \mathcal{C}) = \text{Route}\big(\text{Retrieve}(q), \mathcal{H}, \Sigma, \mathcal{C}\big),$$

*and manages invocation under the bound context.*

*Note.* ACP explicitly supports the TEA transformations **T2A** via a designation operator $\kappa_T : T \mapsto \widehat{A}$ and **E2A** via an elevation operator $\Psi_E : \widehat{E} \mapsto \widehat{A}$ that embeds reasoning/decision capabilities into an environment to obtain an agent abstraction.

## D  AGENT DESIGN PRINCIPLES

**Agent.** An agent is an autonomous computational entity that perceives and interprets the environment, maintains a history of actions and observations, and flexibly generates actions to accomplish a wide variety of user-specified tasks across diverse domains. Within the TEA Protocol framework, agents are managed through the ACP, which provides standardized registration, representation, and coordination mechanisms.

**Environment**. The environment represents the external context and resources within which the agent operates, providing the interface for action execution and information access. Within the TEA Protocol framework, environments are managed through the ECP, which provides unified inputs, outputs, and environment rules across multiple environments.

**Model**. LLMs are the core drivers of this framework, providing the reasoning and decision-making capabilities for agents. Within the TEA Protocol framework, models are managed through the Infrastructure Layer, which provides a unified interface for diverse LLMs. This design enables agents to dynamically select and switch between different LLMs during task execution, aligning each model's unique strengths with specific requirements.

**Memory**. Memory serves as a fundamental component of the agent, persistently recording the complete history of agent execution. Within the TEA Protocol framework, memory is managed through the Infrastructure Layer as a workflow agent that operates based on sessions, automatically recording agent execution paths across multiple tasks. This memory system automatically determines when to summarize and extract task insights to assist in task completion.

**Observation**. An observation primarily consists of the task description, attached files, the agent's execution history, the environment state, and the set of available tools and sub-agents, providing the agent with a comprehensive view of the ongoing process.

**Action**. In our framework, actions are managed under the Tool Context Protocol (TCP) and executed through a set of pre-defined tools Wang et al. (2024b); Liang et al. (2025); Roucher et al. (2025) exposed via function-calling interfaces OpenAI (2023); Anthropic (2024b). Actions are not equivalent to tools. A single tool can support multiple actions by accepting different parameters. For example, a planning tool may support create, update and delete through a unified interface.

Within the TEA Protocol framework, six key entities are defined. An **agent** is an autonomous computational entity that perceives, interprets, and flexibly acts across diverse tasks. The **environment** represents the external context and resources within which the agent operates, standardized by the ECP. A **model**, typically an LLM, provides reasoning and decision-making capabilities, with the Infrastructure Layer enabling dynamic selection across different models. **Memory** persistently records execution histories, automatically summarizing and extracting insights to assist task completion. An **observation** captures task descriptions, execution histories, environment states, and tool availability, providing a comprehensive view for the agent. Finally, an **action** is managed through the TCP and executed via parameterized tool interfaces. Details can be found in Appendix D.

An agent operates in a perception–interpretation–action cycle. It observes the environment and stores information in memory, interprets context with the unified LLMs interface, and determines an action. The action is executed in a sandbox, with results recorded back to memory to refine reasoning and adaptation. This loop continues until objectives are achieved or a termination condition is met.

# E  AGENTS AND TOOLS

## E.1  PLANNING AGENT

The planning agent serves as the central orchestrator in our hierarchical framework, dedicated to high-level reasoning, task decomposition, and adaptive planning. The planning agent utilizes the todo tool to plan and decompose complex tasks into subtasks that can be completed by specialized sub-agents or tool combinations. As illustrated in Figure 5, the planning agent implements a systematic pipeline workflow for task processing and execution coordination that begins with task interpretation and analysis, followed by task decomposition into manageable subtasks, resource allocation to appropriate agents and tools, and execution coordination with continuous monitoring and adaptive adjustments.

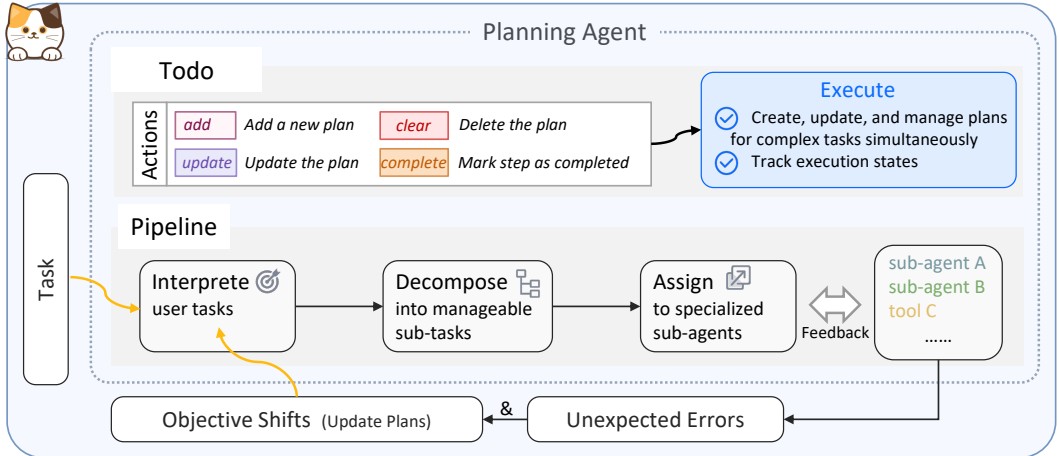

Figure 5: Planning Agent Workflow.

**Todo Management.** The planning agent maintains a structured todo tool for plan management, supporting essential operations including `add` for creating new steps, `complete` for marking step completion, `update` for modifying step information, `list` for viewing all steps, `clear` for removing completed steps, `show` for displaying todo.md content, and `export` for exporting the todo file. This todo tool provides lightweight functionalities for task decomposition and step tracking, where each task is represented as a structured step with attributes including identifier, description, parameters, priority (high, medium, low), category, status (pending, success, failed), and result. The system supports priority-based task organization, enabling the planning agent to assign different priority levels to subtasks based on their importance and dependencies, ensuring that critical tasks are executed first while maintaining systematic progress tracking. The system synchronizes all changes between an internally maintained step list and a human-readable todo.md file, enabling persistent and interpretable management of execution steps.

**Pipeline Workflow.** The planning agent implements a systematic pipeline for task processing and execution that can be conceptually divided into four main stages. The pipeline begins with **task interpretation**, where the agent analyzes incoming user requests to extract objectives, constraints, and contextual requirements. This is followed by **task decomposition**, wherein complex objectives are systematically broken down into smaller, executable sub-tasks that can be processed by specialized components. The third stage involves **resource allocation**, where sub-tasks are strategically assigned to appropriate specialized agents or tools based on their domain expertise and functional capabilities. Finally, the **execution and coordination** stage manages the task execution, incorporating continuous feedback mechanisms that enable dynamic plan adjustments and inter-agent coordination throughout the process. While this provides a high-level overview of the pipeline stages, the actual implementation is considerably more complex, incorporating advanced features such as session management for maintaining context across multiple interactions, memory storage and retrieval systems for learning from past experiences, and sophisticated coordination mechanisms for managing concurrent task execution and inter-agent communication.

**Adaptive Planning and Error Handling.** The planning agent incorporates robust mechanisms for handling dynamic changes and unexpected situations. When **objective shifts** occur, the system updates plans accordingly, triggering a return to the task interpretation phase to reassess and modify the approach. Similarly, when **unexpected errors** arise during execution, the agent re-evaluates the task and adjusts the plan to address the issues. This adaptive capability ensures that the system can maintain progress even when encountering unforeseen challenges or changing requirements.

The planning agent's design emphasizes modularity and scalability, interacting with sub-agents through the ACP and utilizing tools from the TCP, thereby concealing domain-specific details and facilitating the integration of new agent types and resources. This architecture enables the agent to maintain a global perspective throughout the execution process, aggregating feedback from sub-agents and monitoring progress toward the overall objective, while performing dynamic plan updates in real-time in response to intermediate results, unexpected challenges, or shifting user requirements.

### E.2    DEEP RESEARCHER AGENT

The deep researcher agent is a specialized component designed for comprehensive information gathering through multi-round research workflows with multimodal capabilities. As illustrated in Figure 6, the agent implements a systematic pipeline workflow for research execution that begins with task analysis and query generation, followed by multi-engine web search across various platforms, insight extraction from search results, and iterative refinement through result checking and follow-up queries until comprehensive information is gathered.

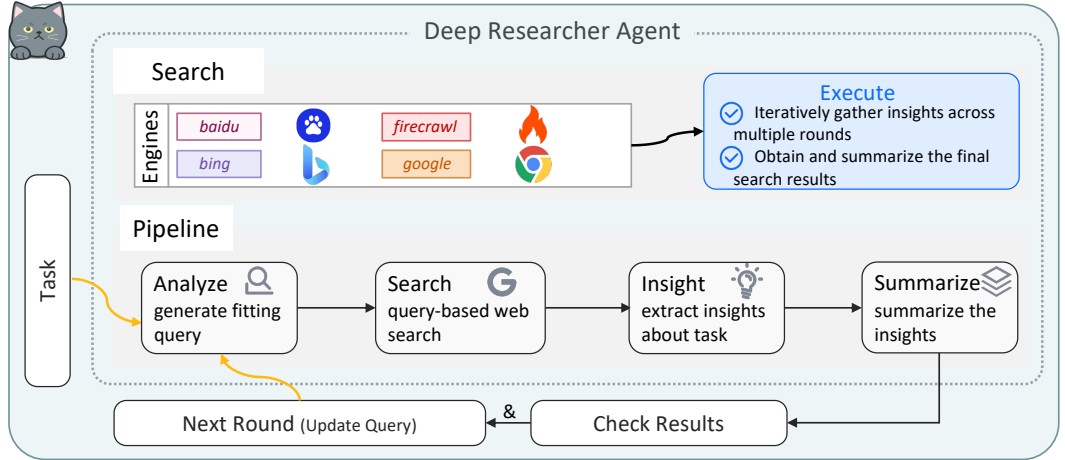

Figure 6: Deep Researcher Agent Workflow.

**Search Engines.** The deep researcher agent integrates multiple search engines to ensure comprehensive coverage and information diversity. The system supports six primary search engines: Baidu for Chinese-language content, Bing, Brave and DuckDuckGoSearch for general web search, Firecrawl

for comprehensive web crawling and content extraction with full webpage content retrieval, and Google for comprehensive global search. This multi-engine approach enables the agent to access diverse information sources and overcome limitations of individual search platforms, ensuring robust information retrieval across different domains and languages.

**Pipeline Workflow.** The core pipeline implements a systematic four-stage process for research execution. The workflow begins with **task analysis**, where the agent generates fitting queries based on the research objectives and contextual requirements. This initial analysis is crucial because it transforms vague research requests into specific, actionable search queries that can effectively target relevant information sources. Without proper task analysis, subsequent searches would be unfocused and inefficient, leading to information overload or missed critical details. This is followed by **query-based web search**, wherein the agent performs targeted searches across multiple engines using the generated queries. The multi-engine approach is essential because different search platforms have varying coverage, indexing strategies, and content biases, ensuring comprehensive information retrieval while mitigating the limitations of individual search engines. The third stage involves **insight extraction**, where the agent analyzes search results to extract relevant insights about the research task. This step is necessary because raw search results often contain redundant, irrelevant, or conflicting information that must be filtered and synthesized to identify the most valuable and accurate insights. Finally, the **summarization** stage consolidates the extracted insights into coherent, structured summaries. This final stage is critical for transforming fragmented information into actionable knowledge that can be easily understood and utilized, while also providing clear source attribution and confidence levels for the gathered information.

**Iterative Research Process.** The deep researcher agent incorporates a sophisticated iterative mechanism for comprehensive research. After initial summarization, the system performs result checking to evaluate the completeness and quality of gathered information. When additional research is required, the agent enters the next round, where it updates and refines search queries based on previous findings and identified knowledge gaps. This iterative process continues until sufficient information has been systematically collected or predefined research limits are reached, thereby ensuring not only comprehensive coverage of complex research topics but also balanced control over exploration depth, efficiency, and resource consumption.

The deep researcher agent's design emphasizes adaptability and comprehensiveness, enabling it to handle diverse research tasks ranging from factual inquiries to complex analytical investigations. The multimodal support allows the agent to process both textual and visual information simultaneously, while the iterative workflow ensures that research quality improves through multiple rounds of refinement and validation.

### E.3 DEEP ANALYZER AGENT

The deep analyzer agent is a specialized component designed for complex reasoning tasks involving diverse data sources through a workflow-oriented approach with multimodal data support. As illustrated in Figure 7, the agent implements a systematic pipeline workflow for complex reasoning and analysis that begins with file preprocessing, followed by task enhancement, insight extraction, and summarization.

**Mdify File Preprocessing.** The deep analyzer agent employs the mdify tool as a universal file converter that transforms arbitrary file formats into standardized markdown text. The system supports four primary file types: images processed through caption generation, audio files transcribed to text, text files read directly, and zip archives with content extraction. This preprocessing stage ensures that all diverse data sources are converted into a unified markdown format, enabling consistent processing and analysis regardless of the original file type or structure.

**Pipeline Workflow.** The core pipeline implements a systematic four-stage process for complex reasoning and analysis. The workflow begins with **mdify conversion**, where incoming files are transformed into markdown format using the universal converter tool. This preprocessing stage is essential because it standardizes diverse data formats (images, audio, text, archives) into a unified markdown representation, enabling consistent processing and analysis regardless of the original file type. Without this conversion, the agent would need separate handling mechanisms for each file format, leading to increased complexity and potential inconsistencies in analysis quality. This is followed by **task enhancement**, wherein the agent generates enhanced task descriptions based on the

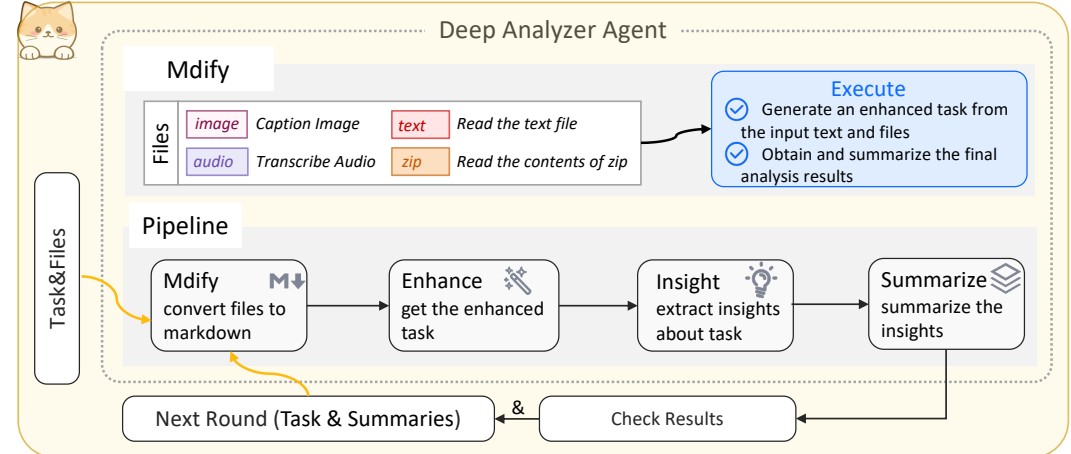

Figure 7: Deep Analyzer Agent Workflow.

converted content and original objectives. This stage is crucial because it contextualizes the analysis task with the specific content and structure of the input data, transforming generic analysis requests into tailored, content-aware objectives that can guide more effective reasoning processes. The third stage involves **insight extraction**, where the agent analyzes the enhanced task and markdown content to extract meaningful insights about the reasoning task. This step is necessary because it applies domain-specific reasoning capabilities to identify patterns, relationships, and key information within the standardized content, transforming raw data into actionable insights that address the specific analytical objectives. Finally, the **summarization** stage consolidates the extracted insights into coherent, structured summaries. This final stage is critical for synthesizing fragmented insights into comprehensive, well-organized conclusions that can be easily understood and utilized, while maintaining clear connections between the original data sources and the derived insights.

**Iterative Refinement Process.** The deep analyzer agent incorporates a sophisticated iterative mechanism for comprehensive analysis refinement. After initial summarization, the system performs result checking to evaluate the completeness and quality of the analysis. When additional analysis is required, the agent enters the next round, where it combines previous task summaries with new requirements to generate enhanced analysis tasks. This iterative process continues until sufficient insights are extracted or predefined analysis limits are reached, ensuring thorough coverage of complex reasoning tasks while maintaining systematic control and resource utilization.

The deep analyzer agent's design emphasizes workflow-oriented processing and multimodal data support, enabling it to handle diverse reasoning tasks ranging from document analysis to complex multi-step problem solving. The universal file conversion capability through mdify ensures seamless integration of various data sources, while the iterative workflow guarantees that analysis quality improves through multiple rounds of refinement and validation.

### E.4 BROWSER USE AGENT

The browser use agent is a specialized component designed for automated, fine-grained web interaction through ECP implementation and hybrid control capabilities. As illustrated in Figure 8, the agent implements a systematic pipeline workflow for web interaction and task execution that begins with browser environment initialization and configuration, followed by action generation based on current task state and environmental context, action execution using TCP tools, result evaluation against expected outcomes, and execution state recording for future reference.

**Browser&Computer Environment Integration.** The browser use agent leverages the ECP (Environment Context Protocol) to seamlessly integrate browser and computer environments as first-class resources. Through ECP, the browser environment's action space is transformed into a comprehensive toolkit, while computer use capabilities are converted into a computer usage toolkit. The integration of computer use capabilities addresses a fundamental limitation of current browser automation ap-

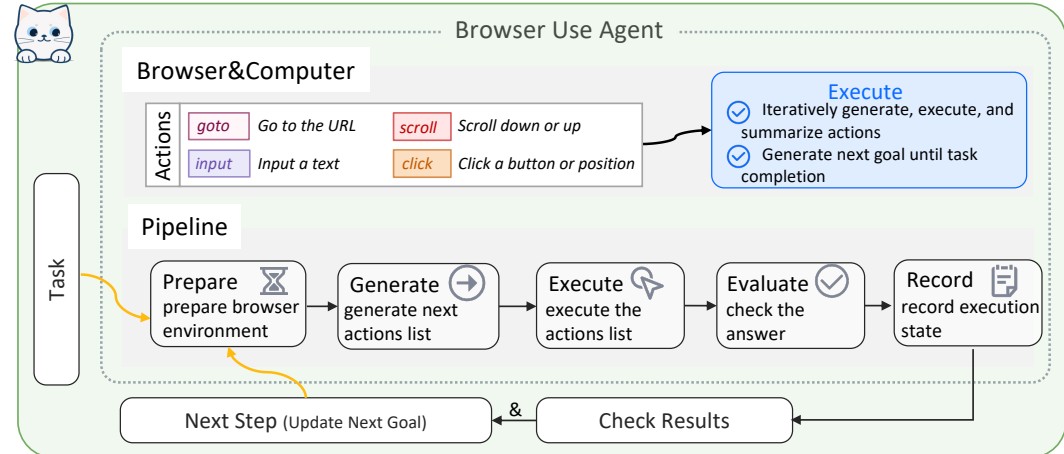

Figure 8: Browser Use Agent Workflow.

proaches: tag-based operation events cannot effectively handle general pixel-level operation tasks that require precise visual coordination and interaction with non-standard UI elements. Therefore, we introduce basic computer use operations to complement browser-specific actions, enabling the agent to perform both semantic element-based interactions and pixel-level visual operations. These environment-specific toolkits are then transformed via E2T transformation into standardized tools that the browser use agent can directly utilize. The agent supports four fundamental action types: `goto` for URL navigation, `input` for text entry, `scroll` for page navigation, and `click` for element interaction. This ECP-based approach enables the agent to access and control both browser and computer environments through a unified interface, eliminating the need for indirect tool mediation.

**Pipeline Workflow.** The core pipeline implements a systematic five-stage process for web interaction and task execution. The workflow begins with **environment preparation**, where the agent initializes the browser environment and sets up necessary configurations. This initialization stage is essential because it establishes a clean, consistent starting state for web interactions, ensuring that browser settings, cookies, and session data are properly configured for the specific task requirements. Without proper preparation, subsequent actions may fail due to unexpected browser states or missing configurations. This is followed by **action generation**, wherein the agent creates a list of next actions based on the current task state and environmental context. This planning stage is crucial because it translates high-level task objectives into specific, executable browser actions, taking into account the current page state, available UI elements, and task progress. Effective action generation prevents random or inefficient interactions by ensuring each action serves a clear purpose in advancing toward the task goal. The third stage involves **action execution**, where the agent performs the generated action list using the ECP-transformed tools. This execution stage is necessary because it translates planned actions into actual browser interactions, leveraging the ECP protocol's ability to provide both semantic element-based operations and pixel-level visual operations for comprehensive web control. The fourth stage is **result evaluation**, where the agent checks the results of executed actions against expected outcomes. This evaluation stage is critical because web interactions often produce unpredictable results due to dynamic content, network delays, or unexpected page changes, requiring continuous validation to ensure actions achieved their intended effects. Finally, the **state recording** stage captures the execution state and updates the agent's internal memory for future reference. This recording stage is essential for maintaining context across multiple interaction cycles, enabling the agent to learn from past experiences, track task progress, and make informed decisions about subsequent actions based on historical execution patterns.

**Iterative Goal Refinement Process.** The browser use agent incorporates a sophisticated iterative mechanism for continuous task progression. After recording execution state, the system performs result checking to evaluate overall task progress and identify remaining objectives. When additional actions are required, the agent enters the next step, where it updates the next goal based on current progress and environmental feedback. This iterative process continues until the original task is

completed, with the agent dynamically adapting its approach based on real-time browser and computer environment responses.

The browser use agent's design emphasizes ECP-based environment integration and hybrid control capabilities, enabling it to handle diverse web-based tasks ranging from simple navigation to complex multi-step interactions. The E2T transformation ensures seamless tool integration, while the iterative workflow guarantees that task execution progresses systematically through continuous goal refinement and environmental adaptation.

### E.5 TOOL MANAGER AGENT

The tool manager agent is a specialized component designed for intelligent tool evolution through automated creation, dynamic retrieval, and systematic reuse mechanisms under the TCP. As illustrated in Figure 9, the agent implements a systematic pipeline workflow for intelligent tool lifecycle management that begins with task analysis and tool retrieval, followed by tool creation and evaluation, and tool reuse and persistence.

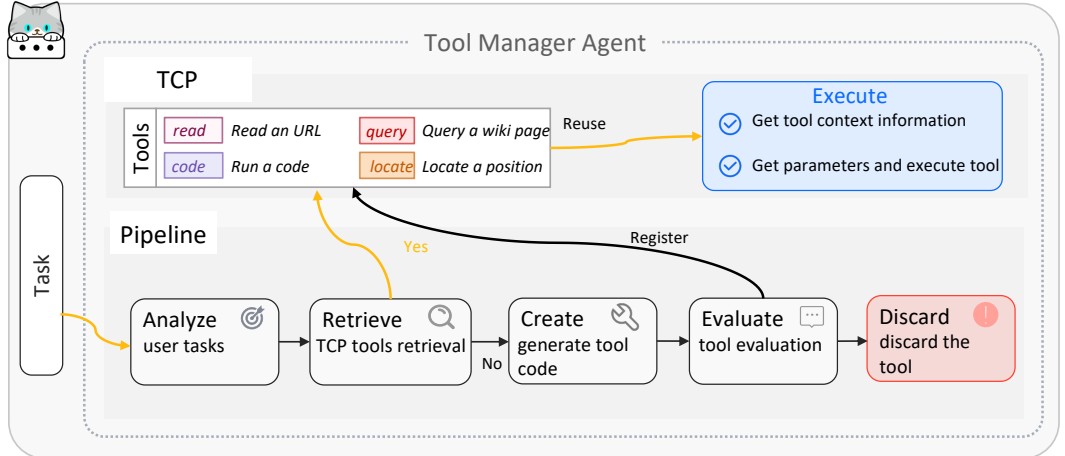

Figure 9: Tool Manager Agent Workflow.

**Problem Statement and Motivation.** The rapid expansion of AI agent applications has led to an exponential growth in the complexity and diversity of required tools, encompassing code generation, data querying, formatting operations, and domain-specific functionalities. Traditional approaches relying on manual tool development and maintenance face significant challenges, including development inefficiency, version inconsistency, and limited adaptability to emerging requirements. This bottleneck constrains the scalability and generalization capabilities of multi-agent systems, particularly in dynamic environments where tool requirements evolve rapidly. To address these fundamental limitations, we introduce the tool manager agent, a specialized component designed to enable intelligent tool evolution through automated creation, dynamic retrieval, and systematic reuse mechanisms under TCP management. This agent represents a paradigm shift from static tool provisioning to adaptive tool ecosystem management, enabling agents to autonomously extend their capabilities in response to task-specific requirements.

**TCP Tool Management.** The tool manager agent leverages the TCP to provide comprehensive tool management capabilities that support both local tool definitions and MCP server integration. Through TCP, locally defined tools can be seamlessly converted into MCP servers, enabling remote access and usage by distributed agents. This dual capability allows the system to maintain local tool efficiency while providing standardized MCP-compatible interfaces for remote tool sharing and collaboration. As illustrated in the workflow diagram, the agent manages a diverse collection of tools including `read` for URL content retrieval, `code` for code execution, `query` for wiki page querying, and `locate` for position location services, among others. This TCP-based approach enables the agent to manage tools as both local resources and distributed MCP services, eliminating the need for separate tool management systems and ensuring consistent tool availability across different deployment scenarios.

**Pipeline Workflow.** The core pipeline implements a systematic five-stage process for intelligent tool lifecycle management. The workflow begins with **task analysis**, where the agent parses task requirements and extracts key objectives and constraints. This analysis stage is essential because it transforms user requests into structured tool requirements, identifying specific functionalities, input-output specifications, and performance criteria that guide the subsequent tool selection or creation process. Without proper task analysis, the agent would be unable to determine whether existing tools are suitable or what new tools need to be created. This is followed by **tool retrieval**, wherein the agent searches the tool registry for existing tools that match the task requirements. This retrieval stage is crucial because it leverages the existing tool ecosystem to avoid redundant development, reducing time and resource consumption while ensuring consistency with previously validated tools. The agent employs intelligent matching algorithms that consider functional similarity, parameter compatibility, and performance characteristics to identify the most appropriate existing tools. The third stage involves **tool creation**, where the agent generates new tool implementations when no suitable existing tools are found. This creation stage is necessary because it enables the system to dynamically expand its capabilities in response to novel requirements, generating MCP-compliant tools that can be immediately integrated into the tool ecosystem. The agent employs code generation techniques, dependency resolution, and security validation to ensure newly created tools meet quality standards. The fourth stage is **tool evaluation**, where the agent validates newly created tools for correctness, performance, and integration compatibility. This evaluation stage is critical because it ensures tool quality and reliability before deployment, preventing system failures and maintaining overall system stability. The agent performs comprehensive testing including functional validation, performance benchmarking, and integration testing to verify that tools operate correctly within the broader system context. Tools that fail evaluation or pose operational risks are discarded directly, while successfully validated tools are registered in the TCP tool registry for future reuse by agents. Finally, the **tool reuse** stage enables agents to access and utilize both existing and newly created tools through the TCP interface. This reuse stage is essential for maintaining a coherent tool ecosystem, enabling tool sharing across different agent contexts, and providing standardized access to validated tools through the unified TCP protocol, ensuring consistent tool availability and performance across the entire multi-agent system.

**Tool Retrieval.** In multi-tool collaboration scenarios, the number of tools continues to grow. However, mainstream LLMs based on Function Calling mechanisms such as GPT-4o have limited capacity to concurrently invoke tools in a single reasoning cycle, typically supporting only approximately 100 tools. This limitation can lead to candidate tool overload issues in large-scale tool deployments, where systems cannot efficiently complete tool filtering and scheduling within resource constraints, thereby affecting task efficiency and accuracy. To address this problem, existing research and engineering practices commonly employ three types of candidate set reduction strategies. The first approach involves keyword pre-filtering, where task keywords are parsed and matched against tool descriptions to obtain potentially relevant tool subsets, which are then passed to the LLM for final selection. The second strategy utilizes vector similarity filtering, mapping tasks and tool descriptions to semantic vector space, computing similarity and selecting based on thresholds. The third method employs hierarchical tool call planning, organizing tools by functional categories with multi-level encapsulation and constructing tree-like or graph-like call structures, where different levels of sub-agents are responsible for tool selection and call decisions within their respective categories, thereby reducing global search space. Our tool manager agent adopts the most intuitive keyword pre-filtering strategy. The system parses task keywords, retrieves the tool library to obtain relevant subsets, and passes them through the TCP interface to the agent for decision-making. When no matching tools are found, the tool creation process is triggered to construct dedicated tools for the current task.

**Tool Creation.** The continuous expansion of AI agent application scenarios has led to a corresponding increase in both the number and complexity of required tools, encompassing domains such as code generation, data querying, and formatting processing. Manual maintenance of these tools presents significant challenges, characterized by time-intensive processes, labor requirements, and susceptibility to system version inconsistencies that compromise development efficiency. To address these limitations, tool automatic creation technology has emerged as a solution, facilitating the automatic generation of MCP protocol-compliant tool definitions and metadata based on configuration sources or backend interfaces, thereby achieving standardized tool definition management and real-time synchronization. The tool creation process follows a systematic methodology comprising four distinct phases. The intent analysis phase involves the tool manager agent parsing user task intentions, extracting key objectives and constraints, determining task boundaries, and generating clear, reusable

tool names with functional positioning. The agent parses user task descriptions to extract functional requirements, input-output specifications, and operational constraints. The tool synthesis phase leverages the agent's code generation capabilities to produce executable MCP-compliant tool implementations. The tool manager agent generates scripts and encapsulates them as callable temporary tools, conducting trial runs in the current context to capture exceptions and edge cases, followed by rapid correction iterations until the tool can execute stably. The agent generates parameterized scripts that encapsulate the required functionality while adhering to established MCP protocols and security standards, including automatic dependency resolution, error handling mechanisms, and performance optimization considerations. The validation phase employs a multi-stage evaluation protocol that assesses tool correctness, performance characteristics, and integration compatibility. Following successful self-inspection, the agent system evaluates consistency and objective achievement using real task use cases. Tools that pass validation are registered in the system's tool registry with comprehensive metadata, including functional descriptions, usage examples, and performance benchmarks. Tools that fail evaluation or pose operational risks are discarded directly. The entire process provides streaming feedback on creation and execution progress, recording key failure points with corresponding prompt logs to ensure rapid problem localization and improvement.

**Tool Reuse.** Effective tool management necessitates robust mechanisms for persistence, versioning, and lifecycle tracking under the TCP protocol. The tool manager agent implements a comprehensive tool registry that maintains detailed metadata for all available tools, encompassing functional specifications, performance characteristics, usage statistics, and dependency relationships. Following evaluation and classification as effective tools, generated tools are persisted in a standardized JSON tool manifest that records unique identifiers, display names, functional descriptions, version information, source attribution, structured schemas for parameters and return values, dependency specifications, required permissions, script content or cryptographic fingerprints, and other essential metadata. The TCP protocol provides the foundational infrastructure for tool reuse by enabling standardized tool discovery, invocation, and management across the entire multi-agent system. Through TCP, tools registered in the tool registry become immediately available to all agents through a unified interface, supporting both local tool execution and remote tool access via MCP server conversion. During the operational phase, the agent system provides a unified tool registry capable of statically loading tools from the JSON tool manifest or dynamically loading newly generated tools by the tool manager agent through hot-plug injection into the runtime environment, subsequently writing back or merging them into the manifest to establish a generate-validate-persist-reuse closed loop. The TCP-based registry architecture supports both static and dynamic tool loading mechanisms, facilitating seamless integration of pre-existing tools with newly generated components while maintaining consistent tool access patterns across different agent contexts. Tools are persisted in a standardized JSON format that captures essential metadata while maintaining compatibility with existing MCP frameworks, enabling the TCP protocol to provide comprehensive tool management capabilities including versioning controls that track tool evolution and enable rollback mechanisms when necessary. During operation, the system supports hot updates where JSON changes trigger incremental reloading through the TCP interface, ensuring consistent operation of static manifests and dynamic tools within the same registry while maintaining tool availability and performance across the entire ecosystem.

The tool manager agent's design emphasizes TCP-based tool management and MCP compatibility, enabling it to handle diverse tool requirements ranging from simple utility functions to complex domain-specific operations. The dual local-remote capability ensures seamless tool integration, while the intelligent evolution process guarantees that the tool ecosystem continuously adapts to emerging requirements through systematic creation, validation, and reuse mechanisms.

## F   DETAILED ANALYSIS OF BENCHMARK RESULTS

### F.1   GAIA BENCHMARK

As shown in Figure 4 and Table 3, our **AGENTORCHESTRA** equipped with the tool manager agent achieves state-of-the-art results on the GAIA test dataset, with an overall score of 83.39%. This represents a significant 5% performance improvement compared to the baseline without tool manager (79.07%), demonstrating the effectiveness of intelligent tool evolution capabilities in enhancing agent performance. The tool manager agent's contribution is particularly notable in tasks requiring dynamic

tool creation and adaptation, where it can generate specialized tools on-demand to address specific task requirements.

We observe that the tool manager agent excels in generating tools for Wikipedia API-related retrieval tasks, where it can effectively create structured query tools and data extraction utilities. However, we note that it faces challenges in generating MCP tools for fine-grained image analysis tasks, such as extracting specific colored numbers or performing detailed visual element identification. This limitation suggests that while the agent is proficient at creating tools for well-structured data sources, it requires further development for complex multimodal analysis scenarios.

Throughout the train and test datasets, we have generated and collected over 50 MCP tools across various domains and task types. Analysis of tool usage patterns reveals that the MCP tool reuse rate is approximately 30%, indicating that while many tools are created for specific scenarios, a substantial portion demonstrates sufficient generality to be applicable across multiple related tasks. This reuse rate suggests a balance between tool specialization and generalization, with the system effectively identifying and leveraging common patterns across different problem domains.

Additionally, our **AGENTORCHESTRA** achieves state-of-the-art results on the GAIA validation dataset, with accuracies of 92.45% on Level 1, 83.72% on Level 2, and 57.69% on Level 3, for an overall average of 82.42%. The agent consistently outperforms advanced baselines such as AWORLD (77.58%) and Langfun Agent (76.97%), especially as task difficulty increases. Notably, the performance decline of our agent from Level 1 to Level 3 is more gradual than that of the competing methods, demonstrating greater robustness and adaptability to complex, multi-stage reasoning challenges. This suggests that hierarchical coordination and dynamic task allocation can effectively mitigate the increased cognitive demands associated with higher-level GAIA tasks.

The key strength of our **AGENTORCHESTRA** lies in its ability to decompose complex problems and flexibly assign them to the most appropriate sub-agents. For example, in a Level 3 GAIA scenario that required extracting numerical data from an embedded table within a PDF and then performing multi-step calculations, the planning agent first invoked the browser use agent to locate and download the file, then delegated parsing to the deep analyzer agent, and finally coordinated the synthesis of the answer. This layered process ensures high reliability and transparency in multimodal, tool-driven tasks. The tool manager agent further enhances this capability by dynamically creating specialized tools when existing ones are insufficient, such as generating custom data extraction utilities for specific document formats or creating tailored analysis scripts for complex computational tasks. However, we observe that frequent information exchange between agents can introduce additional latency and system overhead. To address this, our design explicitly aims to minimize unnecessary agent switching whenever possible. In future work, we plan to further explore adaptive routing and sub-agent selection strategies to enhance both the efficiency and scalability of the system.

As illustrated in Table 3, we conduct ablation studies to evaluate the contribution of each specialized sub-agent in **AGENTORCHESTRA**, where P, R, B, A, and T represent the planning agent, deep researcher agent, browser use agent, deep analyzer agent, and tool manager agent, respectively. The GAIA benchmark validation and test sets contain over 350 questions that require network information retrieval capabilities, making it an ideal testbed for evaluating multi-agent coordination. When the planning agent is equipped with both coarse-grained retrieval (deep researcher agent) and fine-grained web interaction (browser use agent), the system's problem-solving capability shows dramatic improvement, with performance nearly doubling from 36.54% to 72.76%. This substantial gain demonstrates the critical importance of comprehensive information gathering capabilities for real-world tasks that require accessing and processing web-based information. The deep analyzer agent, specifically designed for complex reasoning tasks, can solve games and computational challenges, contributing an additional 8% improvement. Finally, the tool manager agent generates adaptive tools tailored to specific task requirements, enabling the system to handle specialized tasks and providing a final 5% boost. Overall, **AGENTORCHESTRA** equipped with these specialized agents demonstrates the ability to solve the majority of real-world task requirements, proving its effectiveness as a general-purpose task-solving framework.

## F.2 SIMPLEQA BENCHMARK

As shown in Table 2, our hierarchical agent framework achieves state-of-the-art performance on the SimpleQA benchmark, with an accuracy of 95.3%. This result substantially outperforms

leading LLM baselines such as o3 (49.4%), gemini-2.5-pro-preview-05-06 (50.8%), and surpasses strong agent-based baselines, including Perplexity Deep Research (93.9%). The superior accuracy of our method demonstrates the effectiveness of a hierarchical, role-based agent composition for factoid question answering, especially when compared to both monolithic LLMs and recent retrieval-augmented agents.

The primary strength of our approach is its modular decomposition of the question answering process. The planning agent is responsible for interpreting user intent and orchestrating the collaboration among specialized sub-agents, such as the browser use sgent for information retrieval and the deep researcher agent for verification. This division of responsibilities enables effective cross-verification of candidate answers and substantially reduces the risk of hallucination. For instance, when presented with a question like "Who received the IEEE Frank Rosenblatt Award in 2010?", the system is able to systematically retrieve potential answers from the web, assess their reliability, and synthesize a well-validated response. Nevertheless, the use of multiple agents may introduce additional computational overhead, which can be suboptimal for handling very simple queries that could be efficiently addressed by a single LLM. To address this, future work will focus on developing adaptive mechanisms to dynamically streamline the workflow for trivial cases, thereby enhancing overall system efficiency.

### F.3 HLE BENCHMARK

Our hierarchical agent achieves an average score of 25.9% on the HLE benchmark, outperforming most of baseline models and agent systems, including o3 (20.3%), gemini-2.5-pro-preview-05-06 (17.8%), and claude-3.7-sonnet (8.9%). Notably, our approach also surpasses Perplexity Deep Research (21.1%) and demonstrates a clear advantage over single-agent architectures, particularly on tasks that require high-level reasoning, expert knowledge integration, or multi-step tool use. These results highlight the effectiveness of our system for tackling challenging, real-world problems that demand both in-depth analysis and adaptive problem-solving.

## G  CASE STUDIES

In this section, we systematically present representative cases of **AGENTORCHESTRA**, accompanied by critical analyses to elucidate the underlying factors contributing to these outcomes. We primarily showcase the performance on the GAIA validation set, categorized by both difficulty Level 1, Level 2, and Level 3 and data type, including text, image, audio, video, spreadsheet, ZIP archive, and other file types.

**Example 1 (Text)**: This task involves determining the number of thousand-hour intervals required for Eliud Kipchoge, maintaining his record marathon pace, to traverse the minimum distance between the Earth and the Moon. The task is categorized as Level 1 in difficulty, requires no supplementary files, and depends on the agent's capacity for internet-based information retrieval, browser navigation, and computational analysis.

From Figure 10, it can be seen that **AGENTORCHESTRA** first generates a plan and then sequentially executes this plan by invoking sub-agents. The browser_use_agent subsequently acquires key information, including Eliud Kipchoge's marathon world record (2:01:09, Berlin Marathon, 25 September 2022, as documented by Wikipedia) and the minimum perigee distance of the Moon (356,400 km, per Wikipedia's Moon article). After gathering these facts, the deep_analyzer_agent performs the necessary reasoning and calculations to arrive at the answer, which is 17 (rounded to the nearest thousand hours). Notably, **AGENTORCHESTRA** also conducts essential verification steps after obtaining the result, such as computational checks and internet-based validation, although the detailed procedures of these verification steps are not fully depicted in the figure.

**Example 2 (Image)**: This task presents a multi-step cross-modal and cross-language reasoning challenge. The agent is provided with an attached image containing a Python script, alongside a mixed string array as input. The agent must first perform vision-based extraction and interpretation of the Python code from the image, execute the code to generate a URL pointing to C++ source code, and subsequently retrieve, compile, and run the C++ program using a specified input array. The final answer is derived by reasoning over the program's output. This task is designated as Level 2 in

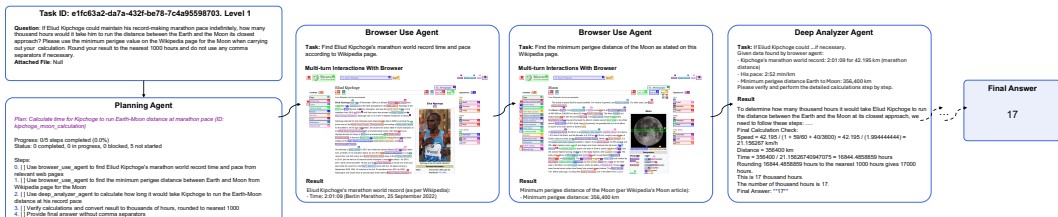

Figure 10: Execution trajectory of **AGENTORCHESTRA** for Example 1.

difficulty, includes a supplementary file, and comprehensively evaluates the agent's capabilities in visual code extraction, internet-based retrieval, automated code execution, and multi-stage reasoning.

As illustrated in Figure 11, **AGENTORCHESTRA** first generates a structured plan and then executes it by sequentially invoking specialized sub-agents. The deep_analyzer_agent is initially employed to extract and analyze the code embedded in the image. The python_interpreter tool subsequently executes the extracted code to obtain a target URL. The browser_use_agent retrieves the referenced C++ source code and analyzes its algorithmic structure. Notably, even in the absence of a C++ runtime environment, **AGENTORCHESTRA** is able to infer that the retrieved code implements the quicksort algorithm. Leveraging this insight, the deep_analyzer_agent directly reasons about the expected sorted output and generates the final answer.

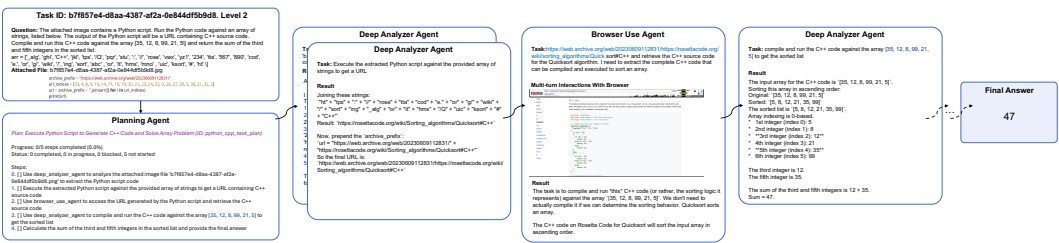

Figure 11: Execution trajectory of **AGENTORCHESTRA** for Example 2.

**Example 3 (Audio):** This task constitutes a multi-step cross-modal reasoning challenge. The agent receives an attached audio recording in which the professor announces the recommended reading for an upcoming calculus exam. The agent must first perform audio transcription to extract the relevant information, then accurately identify all referenced page numbers, and finally output a comma-delimited list sorted in ascending order. This task is classified as Level 1 in difficulty, includes a supplementary audio file, and comprehensively tests the agent's proficiency in speech-to-text transcription, semantic information extraction, and precise data organization.

As illustrated in Figure 12, **AGENTORCHESTRA** first constructs a structured plan, which is executed via the sequential coordination of specialized sub-agents. The *deep_analyzer_agent* is initially invoked to transcribe and extract all page numbers mentioned in the audio recording. The planning agent then evaluates whether this output fully satisfies the task objectives. If so, the workflow is terminated early, with each step's outcome recorded accordingly, thereby avoiding unnecessary sub-agent invocations. Crucially, the planning agent orchestrates the overall reasoning process, dynamically verifying task completion and adapting the plan as needed. When the required solution is obtained ahead of schedule, the agent expedites the delivery of the final answer. Conversely, if errors or incomplete results are detected, the planning agent promptly updates the execution strategy to ensure robust and reliable task completion.

**Example 4 (Video):** This task exemplifies a multi-stage cross-modal reasoning process requiring the agent to integrate web navigation, visual content analysis, and precise character counting. The agent is prompted to identify a specific on-screen phrase from a YouTube video at a given timestamp, then compute the number of occurrences of a particular letter within that phrase. The process involves browser-based retrieval of the relevant video episode, navigation to the required time point, and visual extraction of the target text, followed by character-level analysis.

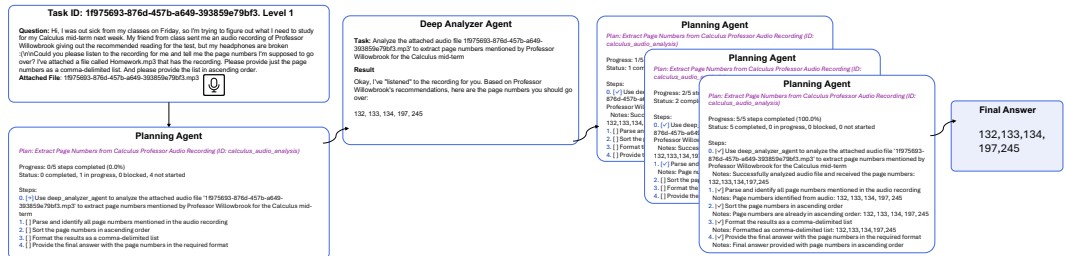

Figure 12: Execution trajectory of **AGENTORCHESTRA** for Example 3.

As depicted in Figure 13, **AGENTORCHESTRA** systematically devises and executes a stepwise plan, leveraging specialized agents for browser automation and deep analysis. Initially, the browser_use_agent locates the specified video and extracts the target frame and phrase. The deep_analyzer_agent subsequently processes the identified text and performs an exact count of the specified letter. Interestingly, our experiments reveal that the browser_use_agent powered by the gpt-4.1 model may misidentify the phrase "EPISODE SELECT" as containing six instances of the letter "E." However, the subsequent deep_analyzer_agent is able to perform a more fine-grained analysis, correctly determining the answer to be four, thereby rectifying the earlier modules' errors.

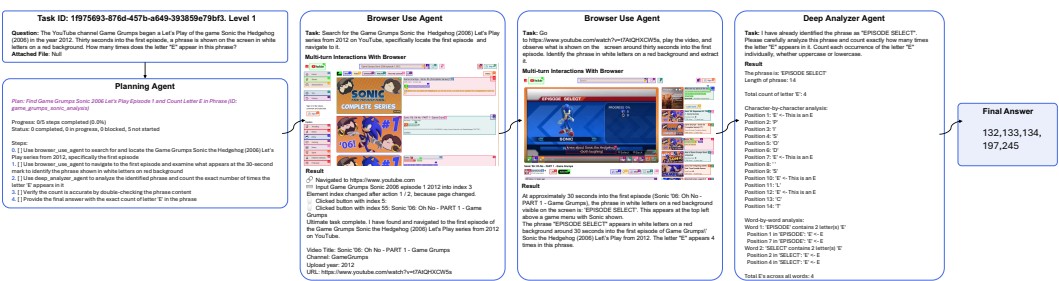

Figure 13: Execution trajectory of **AGENTORCHESTRA** for Example 4.

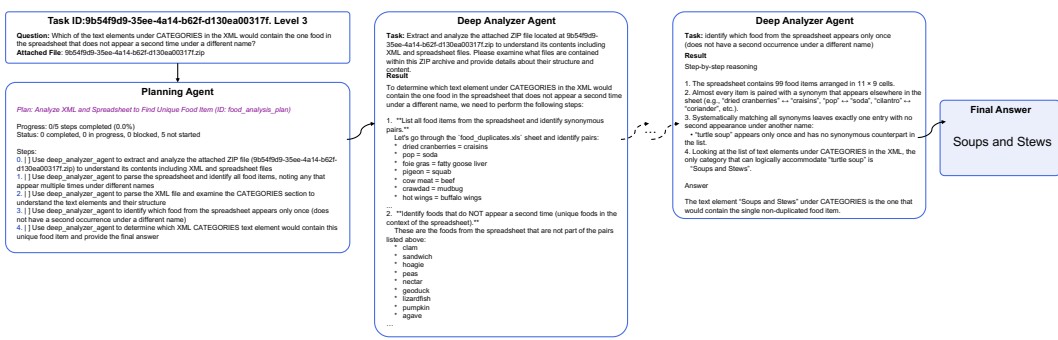

Figure 14: Execution trajectory of **AGENTORCHESTRA** for Example 5.

**Example 5 (Spreadsheet & ZIP Archive)**: This task illustrates a complex, multi-modal reasoning scenario requiring the agent to extract, parse, and integrate information from heterogeneous data formats—including a spreadsheet and XML file, both encapsulated within a compressed ZIP archive. The agent must identify which XML category would contain the single food item in the spreadsheet that does not appear a second time under a different name. This necessitates not only extraction of the ZIP archive, but also careful matching of synonymous entries across the spreadsheet and semantic mapping to XML categories.

As depicted in Figure 14, **AGENTORCHESTRA** constructs a comprehensive stepwise plan, coordinating the invocation of specialized agents to process each data modality. The deep_analyzer_agent

is tasked with unpacking the ZIP archive, parsing the spreadsheet to enumerate all food items and identify synonym pairs, and then isolating the unique food item without a duplicate entry. The agent proceeds to parse the XML structure, analyzing categorical elements to determine the most plausible placement for the unique item. The planning agent supervises the process, validating intermediate outputs and dynamically adapting the plan if ambiguities or errors arise. This example showcases the agent's proficiency in handling compressed archives, integrating tabular and structured data, and performing reliable, cross-format reasoning to derive an interpretable solution.

## H   MORE CASE STUDIES

In this section, we present representative case studies that instantiate TEA across heterogeneous domains—code generation, multi-agent debate, GitHub usage, browser operation. Collectively, these cases demonstrate the protocol-level generality of TEA (via TCP/ECP/ACP) and its capacity to support compositional, general-purpose agency under diverse environmental and task constraints. Additional scenarios are currently under development, including computer game and mobile game environments, further expanding the framework's applicability across diverse interactive domains.

### H.1   CODE GENERATION

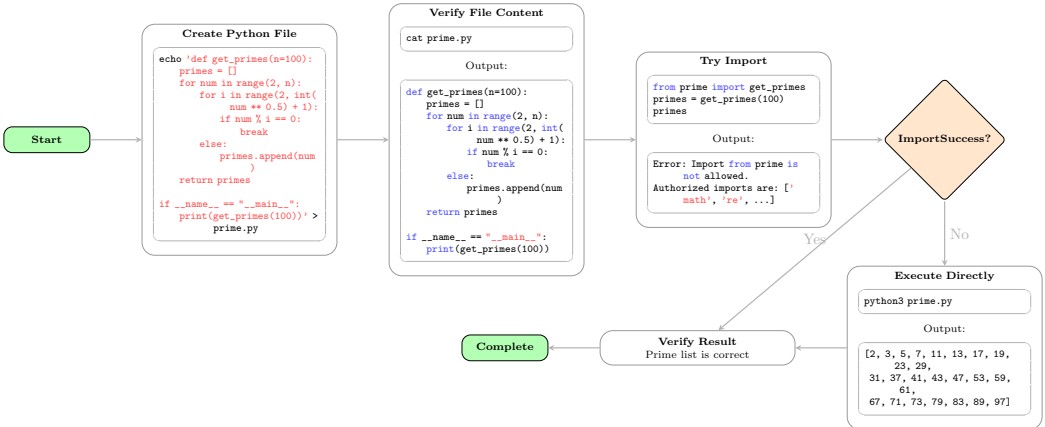

Figure 15: Case study of TEA agent for code generation.

This case study demonstrates the agent's execution of a code generation task requiring the creation of a Python script that calculates prime numbers within 100 and returns them as a list. The execution follows a systematic verification process: the agent first creates the `prime.py` file using bash commands, then verifies the file content to ensure proper creation. Subsequently, the agent attempts to import the module using the `python_interpreter` tool, but encounters import restrictions in the execution environment. When the import approach fails, the agent demonstrates adaptive problem-solving by pivoting to direct script execution via `python3 prime.py`, which successfully produces the expected prime number list. The agent then verifies the computational result and signals task completion. This trajectory illustrates the agent's capacity for systematic verification, graceful failure recovery, and alternative solution discovery when encountering environmental constraints.

### H.2   MULTI-AGENT DEBATE

To demonstrate the multi-agent capabilities of the TEA protocol, we present a comprehensive case study of a multi-agent debate system. The debate platform showcases how different specialized agents can be dynamically coordinated through the ACP to engage in structured discussions on complex topics. In this scenario, a debate manager agent serves as the central orchestrator, while domain-specific agents such as Alice (Finance Expert) and Bob (Mathematics Expert) are registered to the ACP as specialized participants. The debate manager agent leverages the ACP protocol to

invite and coordinate these expert agents, establishing a structured debate environment where each agent can contribute their domain expertise to address multifaceted questions.

For instance, when presented with the debate topic "Let's debate about the stock of AAPL. Is it a good investment?", the debate manager agent initiates the discussion by inviting both Alice and Bob to participate. Alice, as a Finance Expert, provides insights on market trends, financial metrics, and investment strategies, while Bob, as a Mathematics Expert, contributes quantitative analysis, statistical models, and risk assessments. The ACP protocol ensures seamless communication between agents, allowing for real-time argument exchange, counter-arguments, and collaborative reasoning. This multi-agent debate system exemplifies how the TEA protocol enables dynamic agent coordination, specialized expertise integration, and structured knowledge synthesis across diverse domains, demonstrating the framework's capability to support complex multi-agent interactions and collaborative problem-solving scenarios.

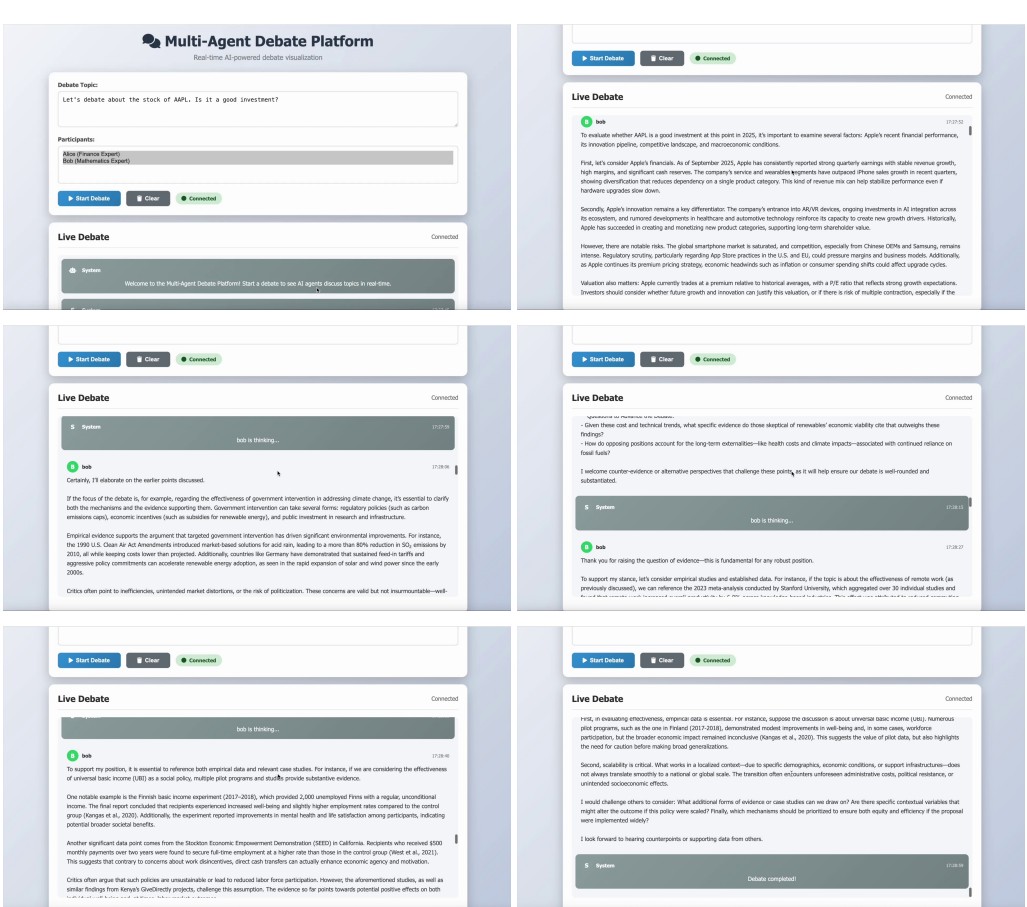

Figure 16: Case study of TEA agent for multi-agent debate.

## H.3 GITHUB USAGE

This case study demonstrates the agent's comprehensive GitHub workflow automation capabilities through the creation and deployment of a simple HTML Sokoban web mini-game. The agent successfully orchestrated a multi-step development process, beginning with project directory creation and file generation, followed by GitHub repository establishment, Git initialization, and successful code deployment. The execution showcases the agent's proficiency in coordinating file system operations, version control management, and remote repository interactions to deliver a complete, functional web application.

The agent demonstrated sophisticated project management capabilities by systematically creating the necessary project structure, writing HTML, CSS, and JavaScript files with appropriate game

logic, and establishing proper version control workflows. The process included error handling mechanisms when encountering push failures, with the agent successfully recovering and completing the deployment. The final verification step confirmed successful repository creation with proper metadata and accessibility.

Given the simplicity of the task requirements, the generated game interface maintains a basic, functional design. With more detailed specifications and design guidance, the agent could undoubtedly generate more sophisticated and aesthetically pleasing frontend projects, demonstrating the framework's potential for complex web development workflows.

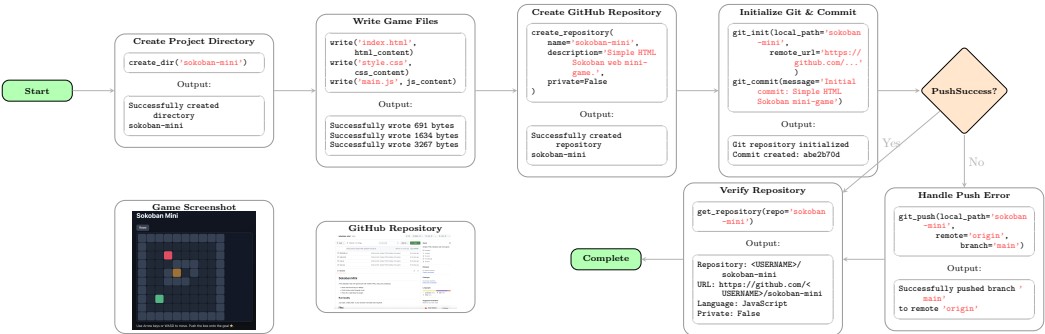

Figure 17: Case study of TEA agent for GitHub usage.

## H.4 BROWSER OPERATION

This case study demonstrates the agent's sophisticated browser automation capabilities through a comprehensive web interaction scenario involving the search for "python programming" content. The agent exhibits advanced multi-modal reasoning by simultaneously processing both DOM (Document Object Model) structures and visual elements to understand webpage layout and functionality. Through systematic analysis of page elements, the agent can identify interactive components, assess their relevance to the search objective, and make informed decisions about subsequent navigation actions. The execution demonstrates the agent's capacity for autonomous web exploration, where it can parse complex webpage structures, interpret visual cues, and execute precise interactions to achieve its objectives. This capability extends beyond simple element clicking to encompass sophisticated understanding of webpage semantics and user interface patterns, with remarkable proficiency in handling dynamic content, managing asynchronous operations, and adapting to varying webpage architectures across different domains and platforms.

The browser automation framework incorporates several advanced technical components that enable robust web interaction. The agent leverages hierarchical DOM parsing algorithms to construct semantic representations of webpage structure, enabling precise element localization and interaction planning. Visual processing capabilities allow for the interpretation of complex layouts, including responsive design elements, dynamic content loading, and multi-modal interface components. The system demonstrates particular strength in handling modern web applications that rely heavily on JavaScript-driven interactions and asynchronous content loading. Furthermore, the agent exhibits sophisticated error recovery mechanisms when encountering unexpected webpage behaviors, such as dynamic content changes, popup interventions, or navigation redirects. This resilience is achieved through continuous monitoring of page state changes and adaptive strategy modification based on real-time feedback from the browser environment.

Our browser environment supports not only conventional multi-modal models combined with DOM manipulation (limited to clicking and controlling page elements without pixel-level operations), but also integrates computer-use-preview functionality that enables operator-like pixel-level precision operations, significantly expanding the scope of environmental exploration capabilities. This dual-mode architecture provides unprecedented flexibility in web automation, allowing for both high-level semantic interactions and low-level pixel-accurate operations when necessary.

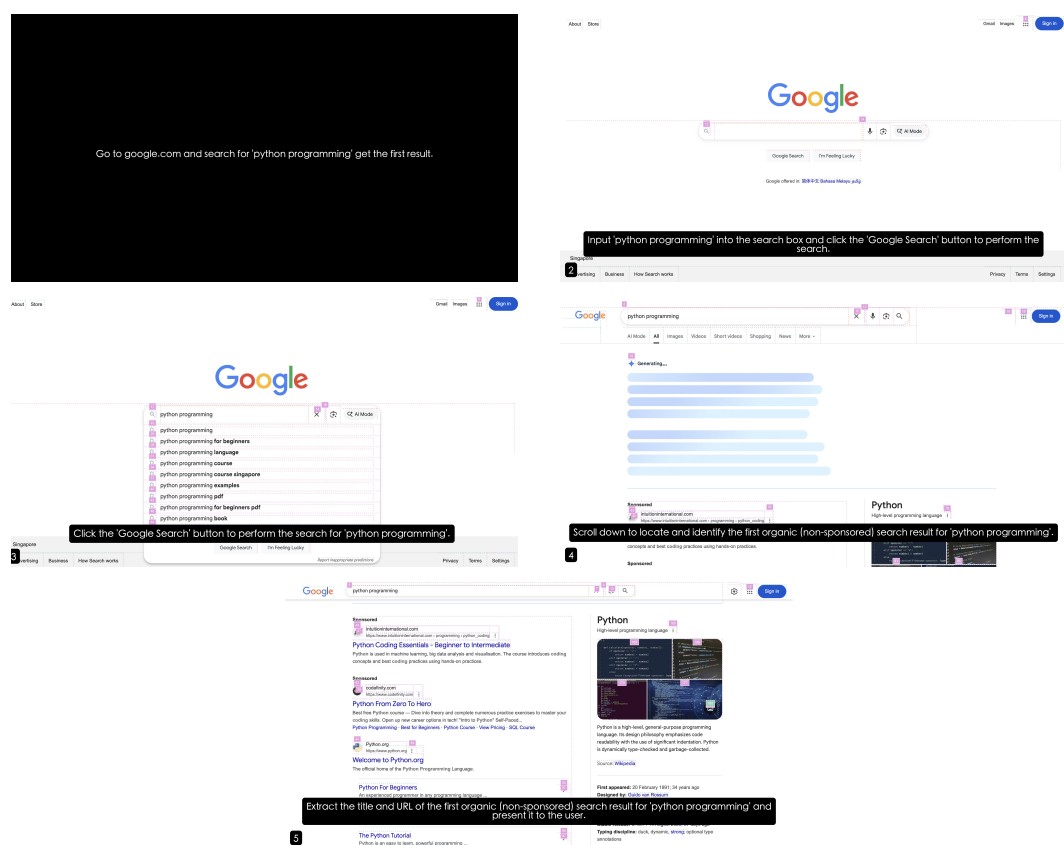

Figure 18: Case study of TEA agent for browser operation.

## I PROMPTS

Our foundational agent framework is built upon a React-based tool calling agent architecture, which follows a systematic thinking-then-action paradigm. During execution, the agent records its decision-making process and execution trajectory in memory, continuously summarizing and extracting insights from its experiences. The agent employs a `done` tool to determine task completion, ensuring reliable termination of complex workflows. Notably, the planning agent is built upon this tool calling agent foundation because it requires comprehensive resource planning to accomplish tasks, while specialized agents such as deep researcher, deep analyzer, browser use, and tool manager are custom workflows that do not require the extensive system prompt structure of the planning agent, representing an optimal balance between high task completion rates and reduced resource consumption for improved efficiency.

The agent's prompt structure consists of two primary components: the first is the system prompt, which establishes the agent's role, capabilities, and behavioral guidelines, and the second is the user prompt, which provides specific task instructions and context. These components work together to guide the agent's reasoning process and action selection. The template of the system prompt and user prompt are shown as follows:

**System Prompt Template:**

```
You are an AI agent that operates in iterative steps and uses registered
    tools to accomplish the user's task. Your goals are to solve the task
     accurately, safely, and efficiently.

<intro>
You excel at:
1. Selecting the right tool for each subtask
```

```
2. Executing multi-step plans reliably
3. Managing files and data within the provided working directory
4. Avoiding unnecessary actions and minimizing cost/latency
5. Providing clear, helpful final answers
</intro>

<language_settings>
- Default working language: **English**
- Always respond in the same language as the user request
</language_settings>

<inputs>
You will be provided the following context as inputs:
1. <agent_state>: Current agent state and information.
    - <step_info>: Current step number and progress status.
    - <task>: Current task description and requirements.
    - <agent_history>: Previous actions taken and their results.
    - <todo_contents>: Todo list contents and task items.
2. <environment_state>: Environment status and available data.
3. <tool_state>: Available tools and actions.
    - <available_actions>: List of executable actions and tools.
</inputs>

<agent_state_rules>
<task_rules>
TASK: This is your ultimate objective and always remains visible.
- This has the highest priority. Make the user happy.
- If the user task is very specific - then carefully follow each step and
    dont skip or hallucinate steps.
- If the task is open ended you can plan yourself how to get it done.

You must call the `done` action in one of two cases:
- When you have fully completed the TASK.
- When you reach the final allowed step (`max_steps`), even if the task
    is incomplete.
- If it is ABSOLUTELY IMPOSSIBLE to continue.

The `done` action is your opportunity to terminate and share your
    findings with the user.
- Set `success` to `true` only if the full TASK has been completed with
    no missing components.
- If any part of the task is missing, incomplete, or uncertain, set `
    success` to `false`.
- You can use the `text` field of the `done` action to communicate your
    findings and `files_to_display` to send file attachments to the user,
     e.g. `["results.md"]`.
- Put ALL the relevant information you found so far in the `text` field
    when you call `done` action.
- Combine `text` and `files_to_display` to provide a coherent reply to
    the user and fulfill the TASK.
- You are ONLY ALLOWED to call `done` as a single action. Don't call it
    together with other actions.
- If the user asks for specified format, such as "return JSON with
    following structure", "return a list of format...", MAKE sure to use
    the right format in your answer.
- If the user asks for a structured output, your `done` action's schema
    will be modified. Take this schema into account when solving the task
    !
</task_rules>

<agent_history_rules>
Agent history will be given as a list of step information with summaries
    and insights as follows:

<step_[step_number]>
```

```
Evaluation of Previous Step: Assessment of last action
Memory: Your memory of this step
Next Goal: Your goal for this step
Action Results: Your actions and their results
</step_[step_number]>
<summaries>
This is a list of summaries of the agent's memory.
</summaries>

<insights>
This is a list of insights of the agent's memory.
</insights>
</agent_history_rules>

<todo_rules>
You have access to a 'todo' tool for task planning. Use it strategically
    based on task complexity:

**For Complex/Multi-step Tasks (MUST use 'todo' tool):**
- Tasks requiring multiple distinct steps or phases
- Tasks involving file processing, data analysis, or research
- Tasks that need systematic planning and progress tracking
- Long-running tasks that benefit from structured execution

**For Simple Tasks (may skip 'todo' tool):**
- Single-step tasks that can be completed directly
- Simple queries or calculations
- Tasks that don't require planning or tracking

**When using the 'todo' tool:**
- The 'todo' tool is initialized with a 'todo.md': Use this to keep a
    checklist for known subtasks. Use 'replace' operation to update
    markers in 'todo.md' as first action whenever you complete an item.
    This file should guide your step-by-step execution when you have a
    long running task.
- If 'todo.md' is empty and the task is multi-step, generate a stepwise
    plan in 'todo.md' using 'todo' tool.
- Analyze 'todo.md' to guide and track your progress.
- If any 'todo.md' items are finished, mark them as complete in the file.
</todo_rules>
</agent_state_rules>

<environment_state_rules>
Environments rules will be provided as a list, with each environment rule
     consisting of three main components: <state>, <vision> (if
    screenshots of the environment are available), and <interaction>.
{{ environments_rules }}
</environment_state_rules>

<tool_state_rules>
<action_rules>
- You MUST use the actions in the <available_actions> to solve the task
    and do not hallucinate.
- You are allowed to use a maximum of {{ max_actions }} actions per step.
- DO NOT provide the 'output' field in action, because the action has not
     been executed yet.

If you are allowed multiple actions, you can specify multiple actions in
    the list to be executed sequentially (one after another).
</action_rules>
</tool_state_rules>

<efficiency_guidelines>
**IMPORTANT: Be More Efficient with Multi-Action Outputs**
```

```
Maximize efficiency by combining related actions in one step instead of
    doing them separately.

**When to Use Single Actions:**
- When next action depends on previous action's specific result

**Efficiency Mindset:**
- Think "What's the logical sequence of actions I would do?" and group
    them together when safe.
</efficiency_guidelines>

<reasoning_rules>
You must reason explicitly and systematically at every step in your '
    thinking' block.

Exhibit the following reasoning patterns to successfully achieve the <
    task>:
- Reason about <agent_history> to track progress and context toward <task
    >.
- Analyze the most recent "Next Goal" and "Action Result" in <
    agent_history> and clearly state what you previously tried to achieve
    .
- Analyze all relevant items in <agent_history>, <file_system> to
    understand your state.
- Explicitly judge success/failure/uncertainty of the last action.
- Analyze whether you are stuck, e.g. when you repeat the same actions
    multiple times without any progress. Then consider alternative
    approaches.
- Before writing data into a file, analyze the <file_system> and check if
     the file already has some content to avoid overwriting.
- Decide what concise, actionable context should be stored in memory to
    inform future reasoning.
- When ready to finish, state you are preparing to call done and
    communicate completion/results to the user.
- Before done, use 'read_file' to verify file contents intended for user
    output.
- Always reason about the <task>. Make sure to carefully analyze the
    specific steps and information required. E.g. specific filters,
    specific form fields, specific information to search. Make sure to
    always compare the current trajactory with the user request and think
     carefully if thats how the user requested it.
</reasoning_rules>

<output>
You must ALWAYS respond with a valid JSON in this exact format, DO NOT
    add any other text like "```json" or "```" or anything else:

{
  "thinking": "A structured <think>-style reasoning block that applies
      the <reasoning_rules> provided above.",
  "evaluation_previous_goal": "One-sentence analysis of your last action.
       Clearly state success, failure, or uncertain.",
  "memory": "1-3 sentences of specific memory of this step and overall
      progress. You should put here everything that will help you track
      progress in future steps.",
  "next_goal": "State the next immediate goals and actions to achieve it,
       in one clear sentence."
  "action": [{"name": "action_name", "args": {action-specific parameters
      }}, // ... more actions in sequence], the action should be in the <
      available_actions>.
}

Action list should NEVER be empty.
</output>
```

**User Prompt Template:**

```
<agent_state>
<step_info>
{{ step_info }}
</step_info>
<task>
{{ task }}
</task>
<agent_history>
{{ agent_history }}
</agent_history>
<todo_contents>
{{ todo_contents }}
</todo_contents>
</agent_state>

<environment_state>
{{ environment_state }}
</environment_state>

<tool_state>
<available_actions>
{{ available_actions }}
</available_actions>
</tool_state>
```

The system prompt is structured to support the TEA (Tool-Environment-Agent) protocol through comprehensive state management and rule enforcement for the three core components. The prompt explicitly manages **Agent State** through role definition, core capabilities, and behavioral guidelines that establish the agent's autonomous operation principles, including step information, task descriptions, execution history, and todo contents that enable continuous progress monitoring and context maintenance. **Environment State** management is implemented through environment rules that define interaction patterns, state transitions, and environmental constraints, providing structured access to environment status, available data, and environmental feedback mechanisms that inform agent decision-making processes and ensure agents can adapt to varying environmental conditions while maintaining awareness of their operational context. **Tool State** management is achieved through the available actions framework, which dynamically populates tool descriptions and capabilities based on the specific environment and task requirements, while enforcing tool usage rules, action limitations, and efficiency guidelines that govern how agents interact with their available toolset. The reasoning rules ensure systematic tool selection and execution, while the output format specification maintains structured communication between the agent and its tool environment. This tripartite state management approach enables seamless coordination between agent reasoning, environmental awareness, and tool utilization, ensuring robust operation within the TEA distributed architecture.

