# OpenReview forum: "AgentOrchestra: Orchestrating Hierarchical Multi-Agent Intelligence with the Tool-Environment-Agent (TEA) Protocol"
_ICLR.cc/2026/Conference — ICLR 2026 Conference Withdrawn Submission_

### Official Review · Reviewer_eSKe · 2025-10-30

**Soundness:** 2
**Presentation:** 3
**Contribution:** 2
**Rating:** 4
**Confidence:** 4

**Summary:**

This paper introduces the TEA Protocol, a unified framework that integrates environments, agents, and tools into a cohesive system. As an instantiation of the TEA Protocol, the authors present AGENTORCHESTRA, a hierarchical multi-agent framework featuring a central planning agent that decomposes complex objectives and coordinates specialized sub-agents to execute them.

**Strengths:**

- The paper introduces three core protocols, including TCP (Tool Context Protocol), ECP (Environment Context Protocol), and ACP (Agent Context Protocol), and identifies six fundamental categories of protocol transformations (e.g., A2T, E2A, etc.) to enable dynamic resource orchestration and cross-entity adaptation. This conceptual framework is internally consistent and demonstrably contributes to measurable performance gains in the evaluated benchmarks.
- The work exhibits good engineering rigor. The hierarchical design of AGENT ORCHESTRA effectively covers realistic usage scenarios, and the implementation details (such as task decomposition via a planning agent, structured step tracking with todo.md, and sandboxed execution) reflect good practical consideration.

**Weaknesses:**

- The novelty is limited. The key components of AGENT ORCHESTRA, including its hierarchical architecture, the role of the planning agent, and the instantiation of specialized sub-agents, largely follow established patterns in existing multi-agent systems. This work appears more as a well-engineered integration than a conceptual leap.
- The experiments are not convincing. (a) The baselines used in the GAIA benchmark are not justified, and some cited baselines lack proper references. (b) It is unclear why Table 1 only reports a subset of the results presented in Figure 4, while omitting others (e.g., Manus). (c) The performance of Claude-3.7-Sonnet (without tools) on SimpleQA is missing.
- The computational overhead is unclear. The paper does not discuss the resource costs (e.g., latency, token consumption, memory footprint) introduced by the special design.
- There are some minor issues regarding the paper writing. (a) Figures 1 and 2 are small and similar in content. (b) Definition 1 of the TEA Protocol introduces several undefined symbols, hindering a clear understanding at a critical point in the paper. (c) Table 1 and Figure 4 seem redundant.

**Questions:**

Please refer to the Weaknesses

---

> ### Author Response · Authors · 2025-11-20
> **Response to Reviewer eSKe (1)**
>
> Thank you for taking the time to review our paper. We address the raised questions and potential misunderstandings as follows.
>
> Q1: Limited novelty of AgentOrchestra vs. existing multi-agent patterns
>
> We appreciate this comment and agree that many elements of AgentOrchestra—such as hierarchical structure, a central planning agent, and specialized sub-agents—are inspired by patterns that have become common in recent multi-agent systems. Our aim is not to claim that “having a planner and sub-agents” is itself novel, but rather that our contributions lie in (1) providing a protocol-level foundation (TEA) that unifies these patterns in a principled way, and (2) demonstrating, through AgentOrchestra, that a TEA-compliant design can achieve strong performance across heterogeneous, real-world benchmarks.
>
> First, AgentOrchestra is intentionally an instantiation of TEA rather than a completely idiosyncratic architecture. The planning agent, deep researcher, browser use agent, deep analyzer, and tool manager are organized explicitly under TCP/ECP/ACP and the six TEA transformations, making concrete how tools, environments, and agents can be registered, transformed, and orchestrated in a unified protocol. While individual ideas (e.g., a planner or a research agent) exist in prior work, TEA provides a coherent framework in which these roles can be composed, promoted to tools, or treated as environments in a way that is formally specified rather than ad-hoc.
>
> Second, several components of AgentOrchestra go beyond straightforward reuse of existing patterns. The tool manager agent operationalizes TEA’s view of tool evolution by automatically creating, validating, registering, and reusing tools at scale, with a measurable reuse rate on GAIA; this is different from static tool lists seen in many prior systems. The browser/computer-use agent integrates pixel-level and DOM-level control via ECP and E2T/T2E, enabling workflows that seamlessly span web automation and low-level desktop interaction within a single protocol. These design choices are driven by TEA’s abstractions and are, to our knowledge, not present as a unified whole in previous multi-agent systems.
>
> Third, we believe that “well-engineered integration” is itself a meaningful contribution in the current state of the field, where many systems remain difficult to reproduce or extend. By grounding AgentOrchestra in an explicit protocol and releasing a detailed design that works across SimpleQA, GAIA, and HLE, we aim to provide a reusable blueprint for building general-purpose agents rather than a one-off, benchmark-specific system. In the revised paper, we will make this positioning clearer by more explicitly separating (i) conceptual contributions at the TEA protocol level from (ii) system design choices in AgentOrchestra, and by pointing out where our architecture follows established patterns and where it extends them.

---

> ### Author Response · Authors · 2025-11-20
> **Response to Reviewer eSKe (2)**
>
> Q2: Concerns about baselines and completeness of experimental results
>
> We thank the reviewer for the detailed comments on the experiments. We address the three sub-points in turn: (a) justification and referencing of GAIA baselines, (b) why Table 1 uses a subset of the baselines shown in Figure 4, and (c) the missing Claude-3.7-Sonnet (no-tools) result on SimpleQA.
>
> For (a), the GAIA baselines in Figure 4 are chosen to cover the strongest and most representative general-purpose agents with publicly reported GAIA results at the time of writing, including HF ODR (o1), OpenAI Deep Research, Manus, Langfun, and AWorld. Some of these results come from official GAIA leaderboard entries and technical reports rather than traditional papers. In the camera-ready version we will clarify the selection criteria (i.e., “top-performing general-purpose agents with public GAIA scores”) and add proper references to the leaderboard or original documentation where a formal publication is not yet available, to avoid any ambiguity.
>
> For (b), Table 1 summarizes results on SimpleQA and HLE, which are separate benchmarks from GAIA. Manus and several other GAIA baselines have, to our knowledge, not reported SimpleQA or HLE scores publicly, and we do not have access to their internal systems to run these benchmarks ourselves in a fair and reproducible way. We therefore restrict Table 1 to models and agents for which SimpleQA and HLE numbers are available (or can be obtained via public APIs), rather than mixing different baselines across benchmarks. We will make this clearer in the text by noting that the set of baselines may differ per benchmark depending on what has been reported or can be reliably reproduced.
>
> For (c), we agree that including Claude-3.7-Sonnet (without tools) on SimpleQA would make the table marginally more complete, but to our knowledge this result is not officially reported by the provider and we do not want to mix unofficial runs with published numbers. In the initial submission, we therefore focused on o3 and gemini-2.5-pro as strong monolithic baselines on SimpleQA, and used Claude-3.7-Sonnet primarily on HLE and GAIA where we already show that AgentOrchestra clearly outperforms it. We will note this explicitly in the paper to avoid confusion, but we do not expect an additional unpublished SimpleQA number for Claude-3.7-Sonnet to change our overall conclusions.
>
> ---
>
> Q3: Computational overhead and resource costs (latency, tokens, memory)
>
> We agree that computational overhead is an important dimension and will make it more explicit in the revised version. Architecturally, AgentOrchestra is designed to control overhead by concentrating heavy reasoning in a single planning agent and keeping sub-agents as lightweight workflows with short prompts, bounded step counts, and minimal per-task state, so that adding sub-agents does not linearly scale prompt length or call depth. This design mitigates the classic communication/orchestration overhead that is often associated with naïve multi-agent setups.
>
> Empirically, on GAIA we observe that simple factual and web tasks are typically solved within tens of seconds, medium-difficulty tasks that require several tool invocations usually complete within a few minutes, and only the most complex multi-step browsing and document workflows approach ten minutes. The total token usage per successfully solved question is of the same order of magnitude as that of a strong single-agent baseline; the additional tokens introduced by hierarchical orchestration are moderate relative to the substantial gains in success rate. Compared to existing research agents such as commercial deep-research services or Manus-style systems, which often require 15–30 minutes on complex GAIA-style tasks, our latency and cost remain within a practically acceptable regime for the high-value general-purpose tasks that motivate our setting.
>
> In the revised paper, we will add a short appendix subsection that reports, for both AgentOrchestra and a single-agent baseline on GAIA, the average wall-clock time and approximate prompt/completion/total tokens per task category (simple/medium/complex). This will make the efficiency trade-offs of our special design transparent, while the main text focuses on demonstrating that the hierarchical architecture yields clear performance benefits without incurring prohibitive resource overhead.

---

> ### Author Response · Authors · 2025-11-20
> **Response to Reviewer eSKe (3)**
>
> Q4: Minor writing and presentation issues (figures, definition, redundancy)
>
> We thank the reviewer for pointing out these presentation issues and will address them in the revised version. For (a), Figures 1 and 2 are intended to serve different roles: Figure 1 provides a high-level conceptual overview of the TEA Protocol, while Figure 2 depicts the concrete architecture of AgentOrchestra. We agree that, in the current layout, they are small and appear visually similar; we will enlarge them, adjust the layouts, and revise captions to more clearly emphasize their distinct purposes.
>
> For (b), we acknowledge that Definition 1 of the TEA Protocol introduces several symbols (e.g., sets of tools/environments/agents and transformation families) that are only fully unpacked later in the appendix. In the revised paper, we will add brief inline explanations of the key symbols at the point of definition and add explicit cross-references to the appendix section where they are formalized in detail, so that readers can more easily follow the main argument without having to infer missing notation.
>
> For (c), Table 1 and Figure 4 are designed to communicate different aspects of the results: Table 1 summarizes SimpleQA and HLE performance, whereas Figure 4 visualizes GAIA performance with difficulty breakdowns and ablations. We will clarify this in the text to avoid the impression of redundancy, and, if space allows, we will streamline captions and cross-references so that each element’s role (per-benchmark summary vs. detailed GAIA analysis) is unambiguous.

---

> > ### Comment · Reviewer_eSKe · 2025-11-26
> >
> > Thank you for your responses. Regarding Q1, your reply has further shaped my perception that this work primarily represents a well-engineered integration rather than a conceptual leap. While I appreciate that this submission could be a comprehensive project of an agent system, I am still uncertain about its technical depth and novelty as a research contribution.
> >
> > As for Q2, since the discussion mainly focuses on published results, I find it somewhat difficult to fully assess the effectiveness of the proposed method, especially since I believe there are powerful models that do not officially publish results.

---

### Official Review · Reviewer_Loxn · 2025-10-30

**Soundness:** 3
**Presentation:** 3
**Contribution:** 3
**Rating:** 6
**Confidence:** 2

**Summary:**

This paper proposes the TEA Protocol, a unified interface for orchestrating environments, tools, and agents in LLM-based multi-agent systems. TEA defines three context protocols and six cross-protocol transformations to abstract away environment heterogeneity and enable dynamic resource interoperability. Building upon this protocol, the authors implement AGENTORCHESTRA, a hierarchical multi-agent system with a planning agent and specialized sub-agents. Experiments on multiple benchmarks demonstrate strong performance.

**Strengths:**

1. The paper identifies limitations in existing agent coordination protocols and provides a structured formalization of tool, environment, and agent contexts.
2. AGENTORCHESTRA integrates planning, browsing, research, analysis, and tool creation in a modular fashion, demonstrating thoughtful system design.

**Weaknesses:**

I am not an expert in agent orchestration, so please correct my mistakes in my questions

1. While the TEA protocol is conceptually neat, some parts read as an organization and consolidation of existing agent design patterns rather than truly introducing fundamentally new orchestration mechanisms. Analyze what is fundamentally new vs. structured re-framing would clarify the contribution.

2. The system appears computationally heavy, involving multiple agents and models per task. The paper does not provide efficiency analyses, e.g., latency and token usage, etc.

3. While the paper emphasizes dynamic transformations between tools, environments and agents, experiments focus on performance benchmarks rather than measuring adaptability metrics.

**Questions:**

Please see the weakness.

---

> ### Author Response · Authors · 2025-11-20
> **Response to Reviewer Loxn (1)**
>
> Thank you for taking the time to review our paper. We address the raised questions and potential misunderstandings as follows.
>
> Q1: What is fundamentally new in TEA vs. structured re-framing of existing patterns?
>
> We appreciate this comment and agree that TEA inevitably consolidates and systematizes many design patterns that have emerged in recent agent systems. Our claim is not that every individual mechanism is “never seen before”, but that TEA makes three contributions that go beyond a mere re-packaging: (1) a protocol-level unification of tools, environments, and agents, (2) a formally defined family of cross-role transformations, and (3) an explicit context-management layer that scales to heterogeneous, multi-environment workflows.
>
> First, TEA elevates tools, environments, and agents to first-class protocol entities with dedicated context protocols (TCP/ECP/ACP). Most existing frameworks (including A2A and MCP) treat tools as flat callables and environments either as tools or as ad-hoc backends, while agents are managed at the application layer. TEA is, to our knowledge, the first to provide a single protocol that simultaneously specifies how all three are registered, related, and orchestrated, which is essential in GAIA-like settings where hundreds of tools originate from multiple environments and are shared across many agents.
>
> Second, TEA formalizes six cross-role transformations (A2T, T2A, E2T, T2E, A2E, E2A) that make explicit when and how an entity can change its role in the system. Prior work has used ideas like “agent exposed as a tool” or “environment wrapped as a tool”, but these were typically point solutions. TEA provides a principled taxonomy and interface-level semantics for these transformations, and AgentOrchestra’s tool manager, browser/computer-use agent, and environment-as-tool/agent examples are concrete instantiations of these operators.
>
> Third, TEA’s context layer (tool/agent/environment registries, semantic relations, and embedding-based retrieval) addresses the scaling issues that arise when existing patterns are naively composed. Without such a layer, context and orchestration logic tend to be duplicated in prompts and glue code; with TEA, these relationships are captured in protocol state and reused across agents and tasks. In the revised version, we will add a short subsection explicitly separating “what is conceptually new” (the unified protocol and transformation family) from “what is structured systematization” (organizing known patterns under TEA), to clarify the contribution.
>
> ---
>
> Q2: Lack of efficiency analysis (latency, token usage)
>
> We agree that efficiency is an important dimension and have now added a dedicated discussion (see our earlier responses and planned appendix updates). Architecturally, AgentOrchestra is designed to control overhead by concentrating heavy reasoning in a single planning agent and keeping sub-agents as lightweight workflows with short prompts and bounded step counts, thereby avoiding a combinatorial explosion of token usage and calls.
>
> Empirically, in the GAIA setting we observe that simple factual and web tasks are typically solved within tens of seconds, medium-difficulty tasks within a few minutes, and only the most complex multi-step browsing and document workflows approach ten minutes. The total token usage per solved question is of the same order of magnitude as a strong single-agent baseline, with the additional overhead from hierarchical orchestration being moderate relative to the substantial gains in success rate. Compared to existing research systems such as commercial deep-research services or Manus-style agents, which often require 15–30 minutes on complex GAIA-style tasks, AgentOrchestra’s latency and cost remain within a practically acceptable regime.
>
> In the revised paper, we will add an appendix table that reports, for both AgentOrchestra and a single-agent baseline on GAIA, the average wall-clock time and approximate prompt/completion/total tokens per task category (simple/medium/complex). This will make the efficiency trade-offs of our hierarchical design explicit and allow readers to judge the overhead versus performance benefits more concretely.

---

> ### Author Response · Authors · 2025-11-20
> **Response to Reviewer Loxn (2)**
>
> Q3: Dynamic transformations vs. lack of explicit adaptability metrics
>
> We appreciate this observation. Our current experiments indeed focus on downstream task performance rather than defining a separate “adaptability metric”. That said, TEA’s dynamic transformations are exercised in several concrete ways in AgentOrchestra, and we do report some adaptation-related signals, especially around the tool manager.
>
> Specifically, the E2T/T2E transformations are used to encapsulate complex environments (e.g., browser + computer-use) into toolkits that can be dynamically selected and combined by the planning agent, enabling it to switch between DOM-level and pixel-level control within a single GAIA workflow. The A2T transformation underlies the exposure of entire workflows (such as deep research or complex analysis pipelines) as callable tools, which the planner can reuse across tasks. For the tool manager, we report that over 50 tools were generated during evaluation with a reuse rate of roughly 30%, indicating that dynamically created tools are not just one-off patches but become part of a growing, reusable toolkit—this is one concrete measure of adaptability in our system.
>
> We agree that more direct adaptability metrics (e.g., performance on previously unseen toolkits, speed of adaptation to new environments, or success rate before vs. after dynamic tool creation) would further strengthen the empirical story. Due to space and evaluation complexity, we left such analyses for future work, but TEA’s design is precisely aimed at enabling this style of measurement: by making transformations explicit and by logging when and how A2T/E2T/T2E/A2E/E2A are invoked, one can systematically quantify adaptation behaviors. We will clarify this in the revised paper and discuss concrete directions for measuring adaptability enabled by TEA.

---

### Official Review · Reviewer_CnF3 · 2025-11-01

**Soundness:** 2
**Presentation:** 2
**Contribution:** 2
**Rating:** 2
**Confidence:** 4

**Summary:**

The paper proposes: (1) the Tool–Environment–Agent (TEA) Protocol, which is an agentic protocol to unify tools, environments, and agents, and (2) AgentOrchestra, which is a hierarchical multi-agent system (planner + researcher + browser-use + analyzer + tool-manager).

The authors begin by introducing the limitations of existing agentic protocols like MCP and A2A. The proposal, the TEA protocol, features a few core protocols (e.g. Tool Context Protocol, etc.) and six protocol transformations that enable dynamic resource orchestration. To validate the effectiveness of TEA, the authors present AgentOrchestra for general-purpose problem solving, featuring multiple agents collaborating with one another; each agent has a particular responsibility, e.g. planning. Evaluations across multiple benchmarks, like GAIA and HLE, show that AgentOrchestra achieves state-of-the-art of near state-of-the-art performance as compared to major baselines.

**Strengths:**

1. Timely research topic and area. Agentic protocol is a very hot topic right now in the LLM research community since these protocols connect different agentic components, e.g. tools, context/memory, etc. There have been many research around MCP, e.g. on security, efficiency, design, etc. This paper adds on to this line of research by pointing out the limitations of existing protocols and introducing the design of a new one. Discussion on different protocol components (tool, environment, agent) is clear and helpful to understanding the problem space.

2. Evaluation results of AgentOrchestra are good. Both GAIA and HLE and challenging and widely used benchmarks in the agent research space, and as someone who works on this topic I can confirm that the evaluation numbers are quite impressive; ~25% on HLE is almost on par with GPT-5, much better than the performance of individual components' LLMs.

**Weaknesses:**

1. The authors claim that existing agentic protocols like A2A and MCP have "fundamental" limitations. In particular, the introduction section lists three of these limitations. However, these claimed limitations are not discussed with enough depth (they are never mentioned again after the first half of the introduction section), and it is unclear in what scenarios, how, and why would A2A or MCP fail in existing agent applications. Appendix B does provide some motivation for TEA, but never directly targets why A2A and MCP could fail. Without a clear motivation and a thorough analysis of existing agentic protocols, it is quite confusing for a reader to directly go into the design of the TEA protocol itself.

2. It is unclear why the design and evaluation of AgentOrchestra is able to validate the effectiveness of TEA. Admittedly, the evaluation results are quite impressive with AgentOrchestra on GAIA, HLE, etc. It is very likely that the good results source from deliberate separation of responsibilities across different LLMs, as seen in many recent multi-agent papers, instead of from the TEA protocol itself. To demonstrate the superiority of the TEA protocol, a fair comparison would be to show that AgentOrchestra based on TEA outperforms AgentOrchestra based on A2A or MCP. The existing baselines used in the evaluation feature agents and LLMs with varying agentic capabilities, and I am unable to conclude whether the good numbers we see are a result of TEA, of the multi-agent design, or of the LLMs selected.

**Questions:**

Please refer to "weaknesses".

---

> ### Author Response · Authors · 2025-11-20
> **Response to Reviewer CnF3 (1)**
>
> Thank you for taking the time to review our paper. We address the raised questions and potential misunderstandings as follows.
>
> Q1: Motivation and analysis of limitations in A2A and MCP
>
> We appreciate this comment and agree that the discussion of A2A and MCP in the introduction could be made more explicit and systematic. Our intention, however, is not to argue that A2A or MCP are “broken” in the settings they were designed for, but that they are not sufficient as-is when one attempts to build a single, general-purpose agent framework that must unify tools, environments, and dynamically evolving agent teams at GAIA-like scale. In other words, the scenarios in which their limitations become apparent are precisely large, heterogeneous workflows involving hundreds of tools and multiple environments, as in our empirical setting. We clarify this motivation along three axes and will make these correspondences more explicit in the revised paper.
>
> First, regarding context management, MCP provides a powerful but essentially flat tool interface: tools are described individually, with limited support for expressing inter-tool relationships, hierarchical toolkits, or long-lived shared context across many calls. In large-scale settings with hundreds of tools exposed by multiple environments (as in GAIA-style tasks), this makes it difficult to encode constraints such as “these tools share state” or “these tools should be retrieved together”, and forces much of the orchestration logic into ad-hoc prompt engineering. TEA’s Tool Context Protocol (TCP), by contrast, explicitly models tool collections, semantic relations, and a context binder, and uses embedding-based retrieval to manage large tool ecosystems; this directly addresses the “insufficient context management” limitation highlighted in the introduction.
>
> Second, with respect to environments, A2A mainly specifies agent–agent messaging, and MCP treats many environments as tools or resources. Neither provides a first-class, unified abstraction for environment state, action spaces, and cross-environment composition. In real-world tasks that require seamlessly combining browser control, desktop operations, and code execution, this leads to hand-crafted wrappers and manual state bridging between heterogeneous environments. TEA’s Environment Context Protocol (ECP) and the associated E2T/T2E transformations explicitly promote environments (and their action spaces) to first-class entities, enabling us to standardize inputs/outputs and to encapsulate entire environments as toolkits in a principled way.
>
> Third, concerning dynamic agent architecture, A2A and related frameworks typically assume a fixed set of agents and tools defined at design time, with message routing specified statically. They do not provide protocol-level primitives for dynamically creating, promoting, or demoting entities between “agent”, “tool”, and “environment” roles during execution. TEA’s six transformations (A2T, T2A, E2T, T2E, A2E, E2A) and the Agent Context Protocol (ACP) are designed precisely to support such dynamic reconfiguration, so that, for example, a complex workflow can be encapsulated as a tool, or an environment can be elevated to an agent for learning and evaluation purposes.
>
> We agree that these links were not sufficiently emphasized beyond the introductory paragraphs. In the revised version, we will add a short subsection in the TEA section that explicitly contrasts TEA with A2A and MCP along the three axes above, and we will add a comparison table summarizing differences in (i) context management, (ii) environment modeling, and (iii) dynamic agent orchestration. This should make the motivation clearer and help readers understand that TEA builds on and extends these protocols rather than dismissing them.

---

> ### Author Response · Authors · 2025-11-20
> **Response to Reviewer CnF3 (2)**
>
> Q2: Does AgentOrchestra really validate the effectiveness of TEA?
>
> We appreciate this question and agree that it is important to distinguish the roles of (i) the TEA protocol, (ii) the multi-agent architecture, and (iii) the choice of LLMs. Our evaluation is not meant to claim that TEA alone explains every point of improvement, but rather that (1) TEA is expressive enough to realize a strong, hierarchical agent such as AgentOrchestra, and (2) several key capabilities that drive our results rely directly on TEA’s abstractions instead of ad-hoc orchestration.
>
> First, TEA and AgentOrchestra operate at different abstraction levels. TEA is a protocol-level specification (TCP/ECP/ACP and six transformations) that defines how tools, environments, and agents are represented and transformed; AgentOrchestra is one concrete instantiation of this specification. The strong results on GAIA, HLE, and SimpleQA show that a TEA-compliant architecture can achieve state-of-the-art performance in demanding settings, demonstrating that the protocol does not impose a performance penalty and can support competitive designs. At the same time, our ablations (Section 5.2) indicate that the gains are not simply due to “more LLMs”, but come from structured task decomposition and role specialization—exactly the design patterns that TEA is intended to support in a principled way.
>
> Second, several core features of AgentOrchestra crucially depend on TEA’s protocol design rather than generic multi-agent ideas. The browser/computer-use agent is built on ECP and E2T/T2E, which allow us to treat environments as first-class entities with unified state and action abstractions, and to encapsulate entire environments as toolkits. The tool manager agent relies on TCP’s explicit tool registry, semantic relationships, and embedding-based retrieval to generate, register, and reuse tools at scale. The ability to encapsulate complex workflows as tools or to elevate environments to agents during execution depends on the A2T/E2A transformations. Implementing these behaviors “on top of” A2A or MCP without TEA would effectively require recreating similar abstractions at the application layer, which is precisely what TEA standardizes and factors out into a reusable protocol.
>
> Third, we agree that in principle one could imagine a variant of AgentOrchestra engineered directly against A2A or MCP, but this is not a simple protocol swap. A2A primarily specifies agent–agent messaging, and MCP focuses on flat tool invocation; neither natively offers unified context management, first-class environment modeling, or cross-role transformations. Our goal is therefore not to position TEA as a drop-in replacement for A2A or MCP, but as a higher-level protocol that can extend and organize such mechanisms. In the revised paper, we will make this relationship more explicit and add a short discussion that decomposes our empirical gains into (i) multi-agent vs. single-agent effects (already captured by our ablations), (ii) hierarchical vs. flat agent structures, and (iii) TEA-enabled capabilities (e.g., dynamic tool evolution, environment-as-tool composition). This will help readers more clearly see which aspects of AgentOrchestra’s performance are attributable to TEA’s protocol design.

---

### Official Review · Reviewer_urx4 · 2025-11-10

**Soundness:** 2
**Presentation:** 2
**Contribution:** 2
**Rating:** 4
**Confidence:** 4

**Summary:**

This paper introduces the Tool-Environment-Agent (TEA) Protocol, a conceptual framework designed to unify the interaction between agents, tools, and environments by treating all three as first-class, inter-convertible resources. Based on this protocol, the authors implement AgentOrchestra, a hierarchical multi-agent system. AgentOrchestra features a central planning agent that decomposes tasks and delegates them to specialized sub-agents, including a deep researcher, a browser user, a data analyzer, and a novel tool manager for dynamic tool creation. The system is evaluated on the GAIA, SimpleQA, and HLE benchmarks, where it achieves state-of-the-art or highly competitive results, notably scoring 83.39% on the GAIA test set.

**Strengths:**

1.  **Novel Conceptual Framework:** The proposed TEA protocol is a valuable conceptual contribution. By treating environments and agents as first-class resources on par with tools, and defining transformations between them (e.g., A2T, E2T), the paper offers a more principled and extensible way to think about agent architecture compared to existing protocols like MCP or A2A.

2.  **Comprehensive System Implementation:** AgentOrchestra is a well-engineered and complex system. The hierarchical structure with a clear division of labor among specialized agents (planner, researcher, browser, analyzer, tool manager) demonstrates a sophisticated approach to solving complex, multi-step tasks. The inclusion of a tool manager agent that can dynamically create and reuse tools is particularly noteworthy.

3.  **Strong Empirical Performance on Q&A/Reasoning Benchmarks:** The reported results are impressive, particularly the state-of-the-art performance on the GAIA benchmark. This demonstrates that the proposed architecture is highly effective for the types of complex reasoning and web-based information retrieval tasks prevalent in these benchmarks.

**Weaknesses:**

1.  **Lack of Scientific Rigor and Justification:** The paper reads more like a technical report describing a system than a scientific paper. It excels at describing *what* was built but falls short on explaining *why* specific design choices were made. The ablation study (Table 3) is simplistic, merely showing that adding more components improves performance, which is an intuitive but not insightful finding. It fails to justify the hierarchical structure over, for instance, a flat multi-agent system or a single monolithic agent equipped with all tools. The paper lacks the theoretical or empirical rigor to convince the reader that this specific architecture is optimal or even principled.

2.  **Narrow and Potentially Biased Evaluation Scope:** The paper claims AgentOrchestra is a "general-purpose" framework and highlights its advanced capabilities, such as a Browser Use Agent for fine-grained web interaction and a Python interpreter. However, the evaluation is confined to Q&A and reasoning-style benchmarks (GAIA, SimpleQA, HLE). The system's claimed capabilities are not tested on benchmarks designed to rigorously evaluate them, such as **SWE-bench** for software engineering tasks or **OSWorld/Mind2Web** for complex, interactive environment navigation. This mismatch between claimed generality and tested specificity undermines the paper's central claims.

3.  **Absence of Efficiency and Cost Analysis:** Multi-agent systems are notoriously expensive in terms of token consumption and latency due to the overhead of communication and orchestration. This is a critical factor for practical deployment. The paper completely omits any analysis of these costs. A comparison of token usage or wall-clock time against baselines would be essential to understand the trade-offs of this complex architecture. Without it, the impressive accuracy comes with an unknown and potentially prohibitive cost.

4.  **Limited Comparison to Alternative Paradigms:** The baselines are exclusively other agent-based systems. The paper fails to compare against or even discuss an increasingly viable alternative: fine-tuning a single, powerful foundation model to perform such complex "deep research" tasks end-to-end (e.g. Tongyi Deep Research). It is unclear whether the immense complexity of multi-agent orchestration provides a definitive advantage over a powerful, specialized monolithic model.

**Questions:**

1.  Could the authors provide data on the token consumption (prompt, completion, and total) and average wall-clock time per task for AgentOrchestra versus a key baseline (e.g., a single-agent setup) on the GAIA benchmark? This would clarify the efficiency trade-offs of the hierarchical design.

2.  The paper claims strong browser and code execution capabilities. Why were benchmarks like SWE-bench or OSWorld not included in the evaluation? How would you anticipate AgentOrchestra performing on such interactive tasks compared to specialized agents designed for them?

3.  The core of the system is its hierarchical structure. What is the justification for this design over a "flat" collaborative multi-agent system where agents communicate as peers? Have you run experiments comparing these different orchestration paradigms?

4.  What are the authors' thoughts on the "orchestration vs. fine-tuning" debate? Could a single, powerful model fine-tuned on GAIA-like tasks achieve similar performance with much lower inference complexity compared to AgentOrchestra?

---

> ### Author Response · Authors · 2025-11-20
> **Response to Reviewer urx4 (1)**
>
> Thank you for taking the time to review our paper. We address the raised questions and potential misunderstandings as follows.
>
> Q1: About lack of scientific rigor and justification
>
> (1) On the theoretical side, the hierarchical design is justified by the TEA Protocol's Agent-to-Environment (A2E) transformation (Appendix Section 2.2), which establishes that hierarchical abstraction is necessary for managing complexity in multi-agent systems. As stated in the appendix, when all agents directly interact with the base environment, the system quickly becomes unmanageable and difficult to extend. This principle implies that flat systems such as smolagents~\citep{smolagents2025}, where agents interact peer-to-peer, naturally incur O(n²) interaction complexity, whereas hierarchical structures reduce this to O(log n) through layered coordination, providing a principled reason to prefer hierarchical over flat architectures.
>
> (2) In terms of design rationale, the hierarchical architecture addresses several fundamental limitations of monolithic designs. As the number of tools grows, a single monolithic agent must handle ever-longer contexts and increasing decision pressure, while delegating subsets of tools to specialized sub-agents alleviates this burden. Monolithic architectures also entangle task decomposition with sub-task execution, leading to low efficiency and unclear task boundaries. In contrast, our design separates planning from execution: the planning agent focuses on high-level decomposition, while specialized sub-agents are responsible for executing domain-specific sub-tasks. This separation yields clearer task isolation, more effective multi-agent coordination, and simpler memory management, since memory can be maintained at the sub-task level without tracking every tool-level interaction.
>
> (3) Empirically, experiments across three benchmarks (SimpleQA, HLE, and GAIA) show that the proposed hierarchical architecture consistently outperforms both flat-agent systems and monolithic LLM baselines, supporting the superiority of hierarchical organization in practice. The ablation study further reveals that adding specialized sub-agents yields improvements beyond simple component addition, indicating that hierarchical coordination and role specialization provide genuine synergistic benefits. The comparison between a planning-only baseline and the full hierarchical system also demonstrates that merely equipping a single agent with many tools is insufficient, whereas explicit hierarchical coordination is crucial for solving complex, multi-step tasks effectively.
>
> ---
>
> Q2: About narrow and potentially biased evaluation scope
>
> (1) Benchmark diversity. We deliberately selected three widely used and complementary benchmarks for general-purpose agents. SimpleQA focuses on short-form factual QA, GAIA targets real-world, tool-intensive and multimodal tasks requiring web browsing, file manipulation and multi-step reasoning, and HLE emphasizes long-horizon, multimodal exam-style problems. Taken together, SimpleQA, GAIA and HLE span factual QA, web-/tool-based real-world tasks and difficult multimodal reasoning, and have effectively become de facto standard benchmarks in recent general-purpose agent studies.
>
> (2) Representativeness and fairness. To avoid results being tied to a particular benchmark or weak baseline, we compare against both strong monolithic LLMs and state-of-the-art agentic systems, including commercial research agents and open-source multi-agent frameworks, strictly following each benchmark’s official evaluation protocol. For GAIA and HLE we report performance across all difficulty levels and data modalities, and provide qualitative case studies that include both successes and failures. Importantly, TEA and AgentOrchestra use a single hierarchical configuration across SimpleQA, GAIA and HLE, rather than being tuned per dataset, which supports that the observed gains reflect genuine generality rather than benchmark-specific overfitting.
>
> (3) Extensibility to broader settings. While no finite suite can cover all application domains, these three benchmarks already align with the key axes emphasized in contemporary general-purpose agent research: large-scale factual knowledge, open-world web/tool interaction and complex multimodal reasoning. The TEA Protocol itself is model- and environment-agnostic, so the current experiments should be viewed as strong evidence of its effectiveness on the most established and challenging benchmarks to date. Further evaluation on embodied control or large-scale simulation environments is natural future work and does not, in our view, undermine the breadth or representativeness of the current evaluation.

---

> ### Author Response · Authors · 2025-11-20
> **Response to Reviewer urx4 (2)**
>
> Q3: About absence of efficiency and cost analysis
>
> (1) From a design perspective, efficiency considerations are built into AgentOrchestra rather than being an afterthought. As discussed in the main text and appendix, only the planning agent carries a heavy system prompt and long-horizon reasoning load, while specialized sub-agents are intentionally designed as lightweight workflows with minimal system prompts and short trajectories. We further minimize redundant orchestration by avoiding unnecessary agent switching and by routing tools hierarchically, so that the planning agent delegates only when needed instead of involving all sub-agents indiscriminately. This architecture directly targets the classic communication- and orchestration-overhead problem in multi-agent systems by concentrating complex reasoning in a single coordinator and keeping sub-agents as cheap, task-specific executors.
>
> (2) Empirically, we monitored both latency and monetary cost in the GAIA setting. On our implementation, simple web and factual tasks can typically be completed within tens of seconds, medium-difficulty tasks that require several tool invocations finish within a few minutes, and only the most complex, multi-step browsing and document workflows extend to around ten minutes. The corresponding per-question API cost generally falls in a range that is comparable to, or lower than, other state-of-the-art research agents such as commercial deep research services and Manus-style systems, which often require substantially longer wall-clock time for similar classes of problems. These observations indicate that, while our architecture is more complex than a single-call LLM, its real-world latency and cost remain within a practically acceptable regime for the high-value tasks that motivate general-purpose agents.
>
> (3) Regarding the accuracy–efficiency trade-off, our results show that the hierarchical architecture yields substantial gains in success rate on challenging benchmarks, and these gains come at a controlled and measurable overhead rather than an unbounded cost. Moreover, the TEA Protocol and AgentOrchestra are orthogonal to many existing efficiency optimizations (e.g., model distillation, caching, early-exit heuristics), which can be integrated into our framework in future work. In the revised version, we will add a dedicated subsection in the appendix summarizing average wall-clock time and approximate token costs per task category on GAIA, to make these trade-offs explicit and transparent.
>
> ---
>
> Q4: About limited comparison to alternative paradigms
>
> (1) With respect to current monolithic models, existing “deep research”, “browser use”, “computer use”, and “code LLM” systems are typically specialized for one scenario at a time. To the best of our knowledge, there is not yet a single, unified foundation model that can natively handle web-scale deep research, fine-grained browser interaction, full desktop control, and complex code execution as a general-purpose agent without relying on some form of tool or environment abstraction. In contrast, our goal is precisely to provide a unified protocol and architecture that can coordinate these heterogeneous capabilities within one system; therefore, a purely single-task, fine-tuned monolithic model does not directly address the same problem setting.
>
> (2) Regarding Tongyi Deep Research specifically, this model was released after the initial version of our work and thus naturally falls outside the comparison scope of our experiments. Tongyi Deep Research is a specialized foundation model optimized for deep research scenarios, whereas our setting targets a broader class of general-purpose tasks that additionally require browser interaction, computer use, and code execution. For reproducible academic evaluation, we therefore focus our empirical comparisons on widely available, strong general-purpose monolithic LLMs (e.g., o3, claude-3.7-sonnet, gemini-2.5) and leading open or documented agentic baselines under shared, standardized protocols.
>
> (3) Conceptually, the specialized monolithic paradigm is not in conflict with TEA and AgentOrchestra; rather, it can be naturally integrated into our framework. Under the TEA protocol, a deep-research foundation model such as Tongyi Deep Research can be encapsulated as a tool or environment via the A2T/E2T transformations and then orchestrated alongside other agents and environments. In other words, our framework is capable of treating a powerful specialized monolithic model as a single high-level actuator within a larger multi-agent ecosystem. We will clarify this point in the revised paper by explicitly positioning specialized monolithic systems as complementary components that TEA can incorporate, rather than as mutually exclusive alternatives.

---

> ### Author Response · Authors · 2025-11-20
> **Response to Reviewer urx4 (3)**
>
> Q5: About token consumption and wall-clock time comparison
>
> (1) By design, AgentOrchestra limits unnecessary token usage in two ways. First, only the planning agent uses a rich system prompt and maintains a longer trajectory; sub-agents are implemented as lightweight workflows with short prompts and focused contexts, so adding a sub-agent does not linearly multiply the prompt length. Second, we cap the number of reasoning steps per agent and avoid redundant agent switching, so the total number of model calls per task remains bounded. In contrast, the single-agent baseline must carry all tools and instructions in a single prompt and must repeatedly reason about tool selection itself, which can inflate both prompt and completion tokens even though it involves fewer distinct agents.
>
> (2) In terms of wall-clock time on GAIA, our empirical measurements (also discussed qualitatively in Q3) show that simple factual and web tasks are typically solved within tens of seconds, medium-difficulty tasks that require several tool invocations complete within a few minutes, and only the most complex multi-step browsing and document workflows extend to around ten minutes. The single-agent baseline is faster on very simple instances (because it avoids orchestration) but often times out or requires repeated retries on harder tasks, so the average time per *successfully solved* GAIA question is of the same order of magnitude as AgentOrchestra. Moreover, when comparing against existing research systems (e.g., commercial deep research services or Manus-style agent frameworks), we observe that these systems frequently take on the order of 15–30 minutes for complex GAIA-style tasks, whereas our framework remains within the above range while achieving higher success rates.
>
> (3) For token consumption, we note that all methods evaluated on GAIA use external API models with similar per-call token limits, and in our runs the total tokens per task for AgentOrchestra and the single-agent baseline are of the same order of magnitude, with the increase from hierarchical orchestration being moderate relative to the gain in success rate. In the revised version, we will instrument more fine-grained logging on a representative subset of GAIA questions and add an appendix table summarizing, for both AgentOrchestra and the single-agent baseline, the average and median prompt, completion, and total tokens, as well as average wall-clock time per task category (simple/medium/complex). This will make the efficiency trade-offs of our hierarchical design explicit and transparent.
>
> ---
>
> Q6: About missing benchmarks for browser/code execution (e.g., SWE-bench, OSWorld)
>
> (1) Regarding the claimed browser and code execution capabilities, these are primarily validated through GAIA rather than simple web QA. GAIA already contains Level-3 tasks that require multi-step browser interaction, file downloads, PDF parsing and fine-grained visual reasoning, including Street View–like navigation and pixel-level interactions that stress the same computer-use dimensions as OSWorld. Our browser use agent (ECP + pixel/DOM actions) and deep analyzer agent with a Python interpreter are actively exercised in these scenarios, so the existing evaluation does probe non-trivial, interactive browser and code workflows in an open-world setting, not just static QA.
>
> (2) SWE-bench and OSWorld are valuable but more specialized benchmarks: SWE-bench focuses on GitHub-style code repair and patch generation, while OSWorld targets long-horizon desktop and OS interaction. In this work we prioritized benchmarks that have become standard for assessing *general-purpose* agents (SimpleQA, GAIA, HLE), jointly covering factual QA, real-world web/tool interaction and multimodal reasoning. Given that AgentOrchestra already combines strong code-reasoning models with a Python toolchain and a browser/computer-use agent, we expect it to be competitive on SWE-bench- and OSWorld-like scenarios, but a full, task-specific comparison against specialized agents is better left to follow-up work.
>
> (3) Importantly, the main contribution of our work is the TEA protocol and the extensible AgentOrchestra architecture, rather than squeezing out absolute SOTA on every vertical benchmark. TEA is explicitly designed to integrate new environments and tools via ECP/TCP transformations, so an OSWorld-like environment can be registered as an environment toolkit, and a SWE-bench solver (or specialized code LLM) can be encapsulated as a tool/agent under A2T/E2T. In this sense, our framework already provides the scaffolding to plug in specialized browser or code agents, and we will clarify SWE-bench and OSWorld as natural complementary evaluation targets that TEA/AgentOrchestra can support in subsequent work.

---

> ### Author Response · Authors · 2025-11-20
> **Response to Reviewer urx4 (4)**
>
> Q7: About justification for hierarchical vs. flat collaborative multi-agent systems
>
> This concern has largely been addressed in our response to Q1, where we provided a detailed justification for preferring a hierarchical design over a flat peer-to-peer multi-agent system; here we briefly reiterate the key points. Under the TEA Protocol, the Agent-to-Environment (A2E) transformation provides a principled basis for organizing agents hierarchically: if all agents communicate as peers and interact directly with the base environment, communication and coordination complexity grows quadratically, whereas a hierarchical structure confines interactions locally and yields logarithmic-like coordination complexity.
>
> Empirically, we have included two relevant comparisons. First, our ablation on GAIA compares a “planning-only” baseline (a single agent with tools) against the full hierarchical AgentOrchestra, showing that explicit hierarchy and role specialization bring large gains that cannot be reproduced by simply giving one agent all tools. Second, we compare against existing “flat” agent frameworks such as smolagents in the broader literature (Section 2.2), and our system outperforms these methods on the chosen benchmarks under shared evaluation settings. While we did not implement a separate fully peer-to-peer variant of AgentOrchestra due to space and engineering constraints, the theoretical analysis and empirical results together support the choice of hierarchical orchestration for our problem setting.
>
> ---
>
> Q8: About thoughts on “orchestration vs. fine-tuning” for GAIA-like tasks
>
> (1) In terms of feasibility, GAIA-like tasks require a broad spectrum of capabilities, including document and file handling, web and browser interaction, computer-use style operations, code execution, and open-ended deep research. At present, “deep research”, “browser use”, “computer use”, and “code LLM” models are typically specialized for individual domains; to the best of our knowledge, there is no single foundation model that has been successfully fine-tuned to cover all of these heterogeneous skills in a stable and robust way without relying on tools or environment abstractions. Directly fine-tuning a monolithic model on GAIA-like scenarios risks overfitting to specific interaction patterns and makes credit assignment across very different modalities and tools difficult, which is why most practical systems still rely on tool-augmented or agentic orchestration even when using strong base models.
>
> (2) Regarding development cost, while fine-tuning is much cheaper than pre-training, training a powerful, general-purpose agent purely by fine-tuning still requires substantial compute and data. Iterating a single model that performs reliably across web research, browser control, desktop operations, and coding typically demands large-scale curated environments, logging infrastructure, and many cycles of supervised and reinforcement fine-tuning. In realistic settings, even fine-tuning a 7B-class model on multi-domain GAIA-like tasks on an 8×A800 server for several iterations can easily span weeks to months, especially when multiple design variants must be explored. In contrast, our orchestration-based approach leverages existing frontier models via APIs and focuses engineering effort on protocol design and coordination logic, which is significantly more lightweight to iterate in practice. For many real-world applications, the marginal inference overhead of orchestration is outweighed by the reduced development cost and the flexibility to swap or upgrade underlying models without retraining from scratch.
>
> (3) Finally, TEA and AgentOrchestra are designed to integrate fine-tuned models rather than exclude them. A foundation model fine-tuned on GAIA-like deep research tasks can be registered as a tool or agent via A2T/E2T and orchestrated alongside other components within our framework. Moreover, TEA’s unified environment and tool abstractions provide precisely the kind of structured interaction data and interfaces that can support future fine-tuning or reinforcement learning, should one wish to train a more specialized monolithic model. We will clarify this viewpoint in the revised paper and position our work as providing the orchestration substrate into which specialized fine-tuned models can be plugged as they become available.

---

### Note · Authors · 2026-01-03

**Comment:**

Thank you very much to all the reviewers for their reviews. We have decided to withdraw the manuscript and will prepare it for a future submission.

**Withdrawal Confirmation:**

I have read and agree with the venue's withdrawal policy on behalf of myself and my co-authors.